# Coronavirus nucleocapsid protein enhances the binding of p-PKCα to RACK1: Implications for inhibition of nucleocytoplasmic trafficking and suppression of the innate immune response

Wenxiang Xue[1], Hongyan Chu[1], Jiehuang Wang[1], Yingjie Sun[1], Xusheng Qiu[1], Cuiping Song[1], Lei Tan[1], Chan Ding[1,2], Ying Liao[1] *

1 Department of Avian Infectious Diseases, Shanghai Veterinary Research Institute, Chinese Academy of Agricultural Sciences, P. R. China, 2 Jiangsu Co-innovation Center for Prevention and Control of Important Animal Infectious Diseases and Zoonoses, Yangzhou University, P. R. China

* liaoying@shvri.ac.cn

**Data Availability Statement:** All relevant data are included in the paper and its Supporting information files. The raw data can be found at the

## Abstract

The hallmark of coronavirus infection lies in its ability to evade host immune defenses, a process intricately linked to the nuclear entry of transcription factors crucial for initiating the expression of antiviral genes. Central to this evasion strategy is the manipulation of the nucleocytoplasmic trafficking system, which serves as an effective target for the virus to modulate the expression of immune response-related genes. In this investigation, we discovered that infection with the infectious bronchitis virus (IBV) dynamically impedes the nuclear translocation of several transcription factors such as IRF3, STAT1, STAT2, NF-κB p65, and the p38 MAPK, leading to compromised transcriptional induction of key antiviral genes such as IFNβ, IFITM3, and IL-8. Further examination revealed that during the infection process, components of the nuclear pore complex (NPC), particularly FG-Nups (such as NUP62, NUP153, NUP42, and TPR), undergo cytosolic dispersion from the nuclear envelope; NUP62 undergoes phosphorylation, and NUP42 exhibits a mobility shift in size. These observations suggest a disruption in nucleocytoplasmic trafficking. Screening efforts identified the IBV nucleocapsid (N) protein as the agent responsible for the cytoplasmic distribution of FG-Nups, subsequently hindering the nuclear entry of transcription factors and suppressing the expression of antiviral genes. Interactome analysis further revealed that the IBV N protein interacts with the scaffold protein RACK1, facilitating the recruitment of activated protein kinase C alpha (p-PKCα) to RACK1 and relocating the p-PKCα-RACK1 complex to the cytoplasm. These observations are conserved across diverse coronaviruses N proteins. Concurrently, the presence of both RACK1 and PKCα/β proved essential for the phosphorylation and cytoplasmic dispersion of NUP62, the suppression of antiviral cytokine expression, and efficient virus replication. These findings unveil a novel, highly effective, and evolutionarily conserved mechanism.

following link. https://zenodo.org/records/
14067323?preview=1&token=eyJhbGciOiJIUzUx
MiJ9.eyJpZCI6Ijk3NGY4Y2IyLWE0M2ItNDI2NS
1iOGVmLTVmNGRmZmZkM2Y4MyIsImRhd
GEiOnt9LCJyYW5kb20iOiIzZGE2NmEwNjlhOT
I4OTIxOGZjYjg1NzNmNzA1NzVlYYj9.
AmQFCLb772-Z8LDAGUVK6llVdt0wNzP7SYGx
2Df52FXjeNkYV4TQxFst2-MmuosNPj4V5AOd8BdI
fx6H9eNDDQ.

**Funding:** This work was supported by the National
Key Research and Development Program
(2021YFD1801104) awarded to Y.L., the National
Natural Science Foundation of China (32372999
and 32172834) awarded to Y.L., and the Shanghai
Natural Science Foundation (23ZR1477000)
awarded to Y.L.. The funders had no role in study
design, data collection and analysis, decision to
publish, or preparation of the manuscript.

**Competing interests:** The authors have declared
that no competing interests exist.

## Author summary

Coronaviruses employ diverse strategies to suppress the host innate immune defense. In
this study, we uncovered a novel and highly effective strategy utilized by diverse coronaviruses to inhibit the innate immune response. Specifically, we found that the coronavirus
N protein facilitates the binding of p-PKCα to RACK1, leading to the phosphorylation of
NUP62 and the cytoplasmic redistribution of multiple FG-Nups. This phenomenon is
accompanied by the disruption of nuclear translocation of several innate immune
response-related transcription factors and suppression of antiviral/pro-inflammatory
gene expression. Our research represents the first elucidation of how the N protein targets
and impairs NPC function through the promotion of p-PKCα- RACK1 interaction and
NUP62 phosphorylation/disassembly. This discovery unveils a novel mechanism
employed by diverse coronaviruses to counteract the host immune response.

## Introduction

The trafficking of proteins and RNA between the nucleus and cytoplasm through NPC is pivotal for numerous cellular functions, such as gene transcription, RNA and ribosomal subunits
export, protein translation, and antiviral innate immunity [1–4]. Comprising the nuclear envelope, NPC, and nuclear transport receptors, the nucleocytoplasmic trafficking system is intricately structured. The outer nuclear membrane (ONM) is contiguous with the endoplasmic
reticulum and shares similarities with it, while the inner nuclear membrane (INM) faces the
nucleoplasm and provides anchoring sites for chromatin and the nuclear lamina [5, 6]. NPC,
acting as the exclusive gateway controlling molecules transport into and out of the nucleus,
consists of approximately 30 different nucleoporins (Nups) with a total molecular mass ~110
MDa [7–9]. Structurally, NPC features an eight-fold symmetric central core surrounding a
transport channel. Nups extending from the central core into the cytoplasm form cytoplasmic
filaments, while nuclear-side filaments interconnect to create the nuclear basket. Within the
central transport channel, Nups with repeating sequences rich in phenylalanine (Phe) and
glycine (Gly), known as FG-Nups, form a cohesive meshwork acting as a permeable barrier to
regulate cargo movement. To cross the nuclear pore, proteins over 40 kDa rely on nuclear
transport receptors such as importin α, importin β or transportins. The small Ras-like GTPase
Ran, which cycles between GDP-bound and GTP-bound states, regulating the formation and
disassembly of nuclear transport receptors with cargo proteins or RNA [10]. The NPC dynamics is directly regulated by cell cycle-dependent phosphorylation [11, 12]. Hyperphosphorylation of the gatekeeper NUP98 and the NUP53 by cyclin-dependent kinase 1 (CDK1) and polo-like kinase 1 (PLK1) is a crucial step promoting NPC disintegration [13, 14]. Several Nups,
including NUP153, NUP214, and NUP358, undergo phosphorylated throughout the cell cycle
and become hyperphosphorylated during M phase, with CDK1 or other kinases likely playing
pivotal roles in this process [15]. Dephosphorylation nuclear envelope proteins by the sequential activated phosphatases is essential for correct NPC and nuclear envelope reassembly. Two
main phosphatases, protein phosphatase 1 (PP1) and protein phosphatase 2A (PP2A), are
involved in the dephosphorylation of Nups and lamins [16, 17]. However, detailed mechanisms underlying the relationship between Nups phosphorylation and NPC disassembly
remain unclear.

The NPC governs the nuclear translocation of key signaling transcription transducers crucial for activating the production of antiviral genes, including IFNs, IFN-stimulated genes

(ISGs), and pro-inflammatory cytokines. The type I IFN signaling pathway constitutes a central component of the antiviral innate immune response, with its activation dependent on the nuclear translocation of IFN regulatory factor 3 (IRF3) to induce the expression of IFNα/β. These type I IFNs subsequently activate the Janus kinase/signal transducer and activator of transcription (JAK/STAT) pathway upon binding to cell surface receptors, facilitating the nuclear translocation of STAT1/STAT2 and ultimately stimulating the expression of numerous antiviral ISGs [18, 19]. In response to infection signals, the transcription factor NF-κB (comprising p50 and p65 subunits) undergoes phosphorylation and translocation into nucleus to induce the expression of various pro-inflammatory cytokines [20]. The phosphorylation and nuclear translocation of p38 MAPK, triggered by stress stimuli or infection, contribute to inflammation by phosphorylating several nuclear transcription regulators and regulating the stress related transcription processes [21, 22]. In addition to importing transcription factors into the nucleus, the NPC serves as the gate for exporting mRNA from the nucleus into the cytoplasm, including mRNA encoding antiviral genes. Consequently, the host nucleocytoplasmic trafficking system represents an effective target for viruses to regulate the host antiviral gene expression and suppress the immune response. Viruses replicating in the cytoplasm mainly employ two strategies to hijack the host nuclear-cytoplasmic transport system and evade antiviral innate immunity. Firstly, viral proteases cleave components of the host nucleocytoplasmic transport system to block the host's innate immune response. For instance, the NS3/4A protease encoded by Hepatitis C Virus triggers the cleavage of importin β1, inhibiting the nuclear transport of IRF3 and NF-κB, thereby disrupting the production of antiviral genes [23]. Although there is evidence that the Dengue Virus and Zika Virus serine protease NS2B3 is involved in Nups cleavage, the impact of this phenomenon on the host innate immune response remains unclear [24]. Secondly, viral proteins directly interact with factors involved in host nucleocytoplasmic trafficking to suppress the host antiviral response. Examples include the Ebola Virus (EBOV) and Japanese Encephalitis Virus (JEV). EBOV VP24 inhibits the interaction of phospho-STAT1 with several importins by binding within the phospho-STAT1 binding region of the importins, ultimately blocking the nuclear import of phospho-STAT1 [25]. Similarly, JEV NS5 blocks the nuclear translocation of IRF3 and NF-κB by competitively binding to importins with IRF3 and NF-κB [26].

Coronaviruses pose a significant global health threat to human beings, leading to substantial economic losses. This RNA virus family possesses a positive-sense, single-stranded genome ranging from 25 to 32 kb. Approximately two-thirds of the 5' genome encode viral replicase polyproteins 1a and 1ab, which are subsequently cleaved into 15–16 mature nonstructural proteins (nsp) crucial for viral replication by internal proteases. The remaining one-third of the 3' genome encodes structural proteins, including spike (S), envelope (E), membrane (M), nucleocapsid (N), and accessory proteins [27]. Highly pathogenic coronaviruses such as SARS-CoV, SARS-CoV-2, and MERS-CoV cause severe atypical pneumonia, accompanied by coughing and high fever, often resulting in high mortality rates. Conversely, mild pathogenic coronaviruses HCoV-229E, HCoV-OC43, HCoV-NL63, and HCoV-HKU1 typically induce common cold symptoms and mild upper respiratory disease [27,28]. Additionally, coronaviruses are also responsible for various infectious diseases in farmed animals. For instance, *gamma*-coronavirus IBV causes highly contagious diseases in chickens, manifesting as bronchitis, nephritis, and fallopian tube injury, leading to a significant decrease in laying rate and chicken production. Since its discovery in the 1930s, IBV has persisted as a significant pathogen, posing a continuous threat to poultry farms [29]. Despite the extensive vaccination effort, controlling this disease remains challenging due to the ongoing emergence of new serotypes and variants [30,31]. Since its discovery in 1971 [32], PEDV has posed a substantial threat as a major pathogen in the pig farming industry, causing clinical symptoms such as vomiting and severe diarrhea. The

mortality rate among infected suckling piglet can reach 100% [33,34]. Coronaviruses have a long history of cross-species transmission, frequently spilling over from wild or domestic animal reservoirs to humans. Zoonotic transmission has been observed in several cases, with three highly pathogenic coronaviruses: SARS-CoV, SARS-CoV-2, and MERS-CoV, believed to have originated from bats, although MERS-CoV used camels as an intermediate host [35–37]. Multigene and complete genome analyses of HCoV-229E, HCoV-OC43, HCoV-NL63, and HCoV-HKU1 also indicate that these coronaviruses have zoonotic origins [38].

Delayed IFN response is a common observation during coronavirus infection, including PEDV [39], SARS-CoV [40], SARS-CoV-2 [41,42], MERS-CoV [43], MHV [44], IBV [45] and PDCoV [46]. Several excellent review articles have extensively discussed how various proteins encoded by different genera of coronaviruses antagonize the innate immune system through diverse strategies [18, 47–49]. Research on targeting the nuclear transport system to suppress innate immunity has largely focused on SARS-CoV ORF6, SARS-CoV-2 ORF6, and MERS-CoV ORF4b. Specifically, SARS-CoV ORF6 interacts with importin α1 (KPNA2) and competes with importin α5 (KPNA1) for binding to importin β1 (KPNB1). This interaction sequesters importin α1 and importin β1 at the ER/Golgi membrane, blocking the nuclear translocation of STAT1 [50]. Similarly, SARS-CoV-2 ORF6 inhibits STAT1/2 nuclear translocation by obstructing the interaction between importin β1 and NUP98, a key nucleoporin involved in the nuclear transport cycle [51]. In the case of MERS-CoV, ORF4b binds to KPNA4 during infection, inhibiting its interaction with the p65 subunit and thereby interfering with p65 nuclear translocation [52]. Both ORF6 and ORF4b are accessory proteins unique to these coronaviruses and are absent in other coronaviruses. Whether there is/are evolutionarily conserved coronavirus protein(s) involved in modulating the nuclear trafficking system remains an open question and warrants further investigation.

In this study, we utilized IBV as the model to explore the shared mechanisms utilized by diverse coronaviruses to disrupt the host nuclear transport system. We observed that IBV infection triggered cytoplasmic dispersion of several FG-Nups and hindered the nuclear ingress of various transcription factors and p38 MAPK, consequently impeding the transcriptional activation of antiviral and pro-inflammatory genes. Further study revealed that the IBV N protein was accountable for the disturbance of nucleocytoplasmic trafficking by enhancing the association of p-PKCα with scaffold protein RACK1, leading to relocation of the p-PKCα-RACK1 complex to cytoplasm. This promoted the phosphorylation and cytoplasmic dispersion of NUP62, ultimately impeding the nuclear import of several transcription factors and dampening the transcription of antiviral/pro-inflammatory genes. Importantly, this novel function was conserved among N proteins from four genera of coronaviruses.

## Results

### IBV infection disrupts the nuclear translocation of transcription factors and antagonizes the expression of antiviral and pro-inflammatory cytokines

The previous report has demonstrated that IBV inhibits IFNβ-mediated nuclear translocation of STAT1 at the late stages of infection [53]. In this study, we explored whether IBV also impedes the nuclear translocation of additional transcription factors, including IRF3, STAT1, STAT2, and p65. Initially, Vero cells, an adapted cell line of the IBV Beaudette strain, were infected with 1 MOI of IBV, followed by poly(I:C) transfection to induce IRF3 nuclear translocation. Western blot analysis showed that both IBV infection and poly(I:C) stimulation induced phosphorylation of IRF3, and IBV infection did not inhibit poly(I:C)-stimulated IRF3 phosphorylation (Fig 1A, bottom panel). However, immunofluorescence analysis depicted in

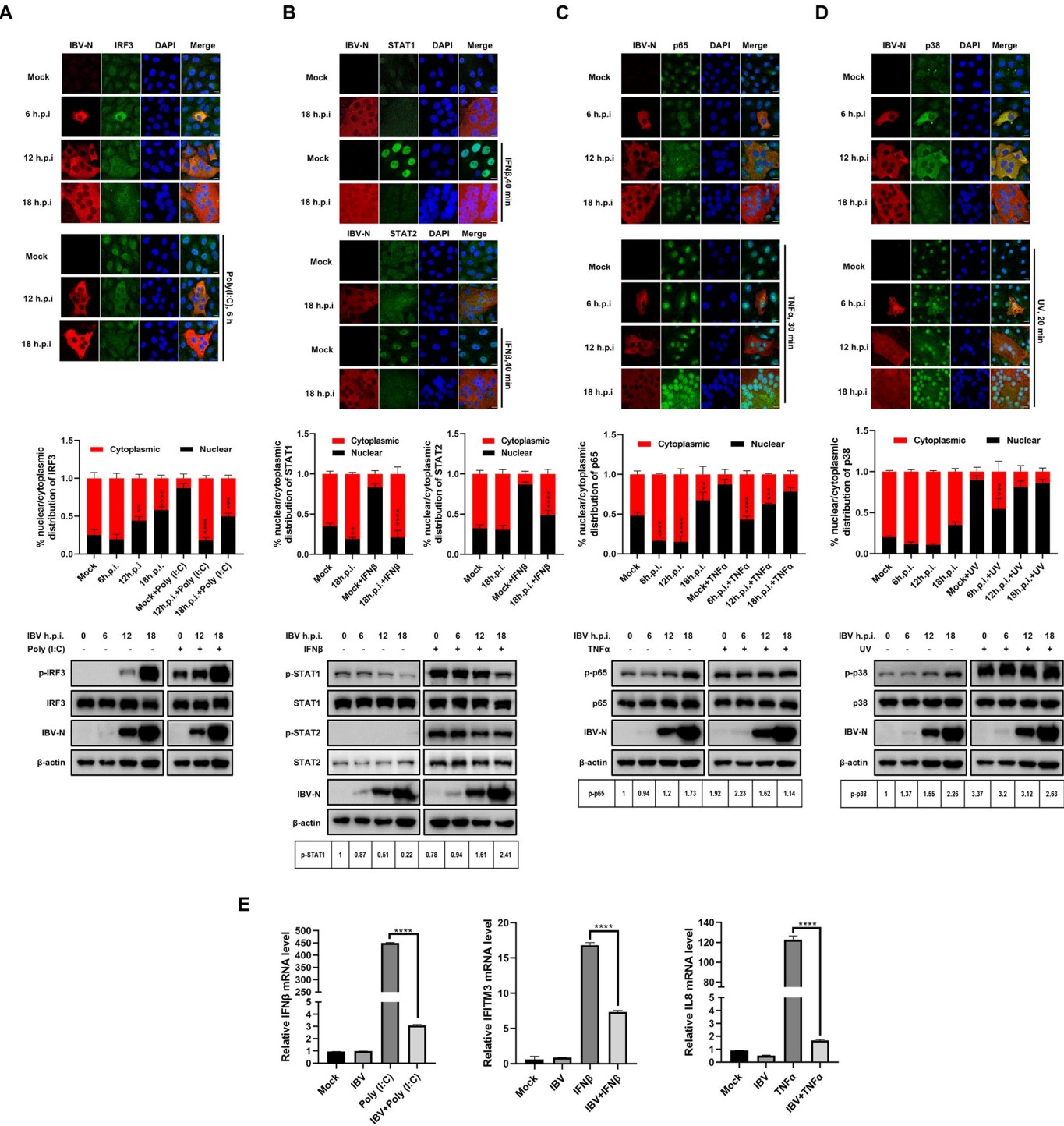

**Fig 1. IBV infection impairs nuclear translocation of transcription factors and suppresses transcription of antiviral genes and pro-inflammatory genes.**
(A-D) Vero cells were infected with IBV at an MOI of 1, followed by transfection with poly(I:C) for 6 h, treatment with IFNβ for 40 min, treatment with TNFα for 30 min, or exposure to UV irradiation (1.92 J/cm2) for 20 min. Mock-infected cells served as the control group. Cells were harvested at the indicated time points and subjected to immunofluorescence analysis and Western blot analysis. Representative images from three independent experiments are shown, with scale bars indicating 10 μm. The fluorescence signals of corresponding proteins (IRF3, STAT1, STAT2, p65, and p38) in mock-infected cells and IBV-infected cells from three fields of view were quantified using ImageJ. The intensities of the fluorescence signals in the nucleus (black bars) and the cytoplasm (red bars) are presented in bar graphs. Error bars represent the standard deviation (SD). (E) DF-1 cells were infected with IBV at an MOI of 5 for 2 h, followed by transfection with poly(I:C) or treatment with TNFα for 6 h. Cells were harvested at 8 h.p.i. and subjected to qRT-PCR analysis. For detection of IFITM3 expression, DF-1 cells were infected with IBV at an MOI of 5, followed by treatment with IFNβ for 6 h at 6 h.p.i. Cells were harvested at 12 h.p.i. and subjected to qRT-PCR analysis. Mock-

infected cells served as the control group. Error bars represent the SD of technical triplicates. Statistical significance levels are denoted as follows: ns (not significant), P > 0.05; *P < 0.05; **P < 0.01; ***P < 0.001; ****P < 0.0001.

Fig 1A (top panel) revealed that throughout the infection process (6–18 h.p.i.), IBV infection failed to induce the nuclear entry of IRF3, whereas poly(I:C) transfection effectively stimulated IRF3 nuclear translocation in mock-infected cells. Intriguingly, in poly(I:C)-treated and IBV-infected cells, IRF3 displayed a cytoplasmic distribution. These results demonstrate that although IBV infection leads to IRF3 phosphorylation, it does not induce its nuclear entry and inhibit poly(I:C)-stimulated IRF3 nuclear entry. This indicates that IBV infection hinders the nuclear transport process of phosphorylated IRF3.

Since Vero cells lack IFN, they are suitable for the study of JAK-STAT pathway [54, 55]. Immunofluorescence analysis demonstrated that IBV infection did not trigger the nuclear translocation of STAT1 and STAT2, whereas IFNβ successfully induced their nuclear translocation. In IBV-infected Vero cells, the nuclear translocation of STAT1 and STAT2 stimulated by IFNβ was impeded, indicated by the diffuse signals of both transcription factors observed in both the cytoplasm and the nucleus (Fig 1B, top panel). Western blot analysis showed IBV infection did not promote the phosphorylation of STAT1 and STAT2 and significantly inhibited IFN-β-induced STAT1 phosphorylation at the late stage of infection (18 h.p.i.) (Fig 1B, bottom panel). This result suggests that IBV not only inhibits the nuclear translocation of STAT1 and STAT2 but also inhibits STAT1 phosphorylation at late infection stage, consistent with previous reports [53].

The inhibition of nuclear translocation of aforementioned transcription factors by IBV infection led us to investigate whether the virus also inhibits the nuclear translocation of additional transcription factors or transcription transducers. Consequently, the nuclear translocation of pro-inflammatory transcription factor NF-κB subunit p65 and p38 MAPK was examined. As illustrated in Fig 1C–1D, IBV infection did not promote the nuclear translocation of p65 and p38 MAPK at 6 h.p.i. and 12 h.p.i.; however, along with the infection progressed, an increasing amount of p65 and p38 MAPK entered the nucleus at 18 h.p.i.. Both TNFα treatment and UV irradiation effectively promoted the nuclear entry of p65 and p38 MAPK, whereas IBV infection led to a proportion of p65 and p38 MAPK remaining in the cytoplasm, despite the detection of intense nuclear signals (Fig 1C–1D, top panel). Western blot analysis showed that IBV infection induced phosphorylation of p65 and p38 MAPK at the late infection stage (18 h.p.i.) (Fig 1C–1D, bottom panel). These findings suggest that the virus suppresses inflammatory transcription events during the early stages of infection to facilitate successful infection; however, inflammation is ultimately induced at the late stage of infection. The inhibition of nuclear translocation of IRF3, STAT2, and p65 by IBV infection was further validated in host chicken embryo fibroblast DF-1 cells (S1A–S1B Fig).

Concurrently, the transcription of IFNβ (induced by IRF3), IFITM3 (an ISG induced by STAT1/2), and pro-inflammatory cytokine IL-8 (induced by NF-κB and p38 MAPK) was suppressed by IBV infection in DF-1 cells, which possesses a complete IFN signaling pathway (Fig 1E). Collectively, these results demonstrate that IBV infection suppresses nuclear entry of multiple transcription factors, particularly those involved in antiviral IFN pathways.

## IBV Infection induces the phosphorylation of NUP62 and perturbs the integrity of NPC

As IBV infection disrupts the nuclear translocation of multiple transcription factors, we hypothesized that the infection likely interferes with the nucleocytoplasmic trafficking. To

investigate this, we analyzed the integrity of NPC in IBV-infected Vero cells by examining the subcellular distribution of FG-Nups using the monoclonal antibody mAb414. This antibody recognizes a number of FG-Nups, including NUP62, NUP153, NUP98, NUP35, NUP54, NUP58, NUP214, and NUP358 [24], which constitute the mesh in the NPC central channel and are crucial for cargo molecules transport [56]. Immunofluorescence analysis revealed that FG-Nups detected by mAb414 were predominantly localized to nuclear envelope as ring signals; however, in IBV-infected cells, the FG-Nups were dispersed into the cytoplasm at 6 h.p.i. and 12 h.p.i.; they were gradually relocated to the nuclear envelope as ring signals from 12 h.p.i. to 18 h.p.i., coinciding with the formation of large syncytia (Fig 2A, top left panel). These observations suggest that IBV dynamically modulates NPC integrity during infection progression. To corroborate these findings, we performed immunostaining of specific Nups with antibodies against NUP62, NUP153, NUP98, NUP42, and TPR. Results indicated that these specific Nups were primarily localized to the nuclear envelope or inside the nucleus; however, in IBV-infected cells, NUP62, NUP153, NUP42, and TPR were redistributed to the cytoplasm at 6 h.p.i. and 12 h.p.i.; while at 18 h.p.i., NUP42 and TPR were relocated to the nucleus or nuclear envelope, whereas NUP153 remained dispersed in the cytoplasm and NUP62 was redistributed to one side of the cytoplasm with intense signals (Fig 2A). Compared to other Nups, only a small proportion of NUP98 was dispersed into cytoplasm at 6 h.p.i.; as infected cells formed syncytia at 12 h.p.i. and 18 h.p.i., NUP98 was relocated to nuclear envelope as ring signal. These observations confirm that multiple Nups are dislocated from the nuclear envelope and the NPC integrity is disrupted during early stages of infection (6–12 h.p.i.); at late stage of infection (18 h.p.i), NPC integrity is not fully recovered, as evidenced by the mislocalization of NUP153 and NUP62. Notably, the cytoplasmic dispersion signals of these Nups colocalized well with IBV N protein at 6 h.p.i. and 12 h.p.i., suggesting potential interaction.

Subsequently, we examined the subcellular localization of soluble phase components of nucleocytoplasmic trafficking system: the key receptors importin α1 and importin β1, as well as Ran, which governs the formation and disassembly of transport receptors with cargo proteins. As depicted in Fig 2B, importin β1 exhibited a ring signal adjacent to the nuclear envelope. However, during IBV infection at 6 h.p.i., a certain proportion of importin β1 was dispersed into the cytosol. Despite this, the majority of this receptor remained associated with the nuclear envelope throughout the infection time course. Importin α1 was predominantly dispersed in the cytoplasm, and IBV infection did not obviously alter its localization. Ran was primarily localized inside the nucleus; however, a small proportion of Ran was translocated to the cytosol and colocalized well with the N protein at 6 h.p.i. and 12 h.p.i. As the infection progressed, Ran relocated to the nucleus at 18 h.p.i..

To validate these results, we further examined the subcellular localization of FG-Nups and Ran in DF-1 cells. Due to the unavailability of antibodies against chicken Nups, only antibodies against human NUP62, NUP98, NUP153, TPR, and Ran, which could cross-react with the corresponding chicken proteins, were applied in this study. Immunofluorescence analysis showed that these proteins were primarily localized to the nucleus or nuclear envelope; however, in IBV-infected cells, certain proportion of NUP62, NUP98, NUP153, TPR, and Ran were redistributed from the nuclear envelope or nucleus to the cytoplasm, colocalizing well with N protein (S2A Fig). These results are consistent with the observations in Vero cells. To rule out the possibility of Nups' cytoplasmic dispersion and their colocalization with the N protein being artifacts resulting from the cross-reaction of two fluorescent secondary antibodies with the same primary antibody, we performed staining for NUP62 and N protein or NUP153 and N protein using a different set of fluorescent secondary antibodies in reverse. The result showed that NUP62 and NUP153 were still dispersed into cytoplasm and colocalized perfectly with IBV N protein (S2B Fig).

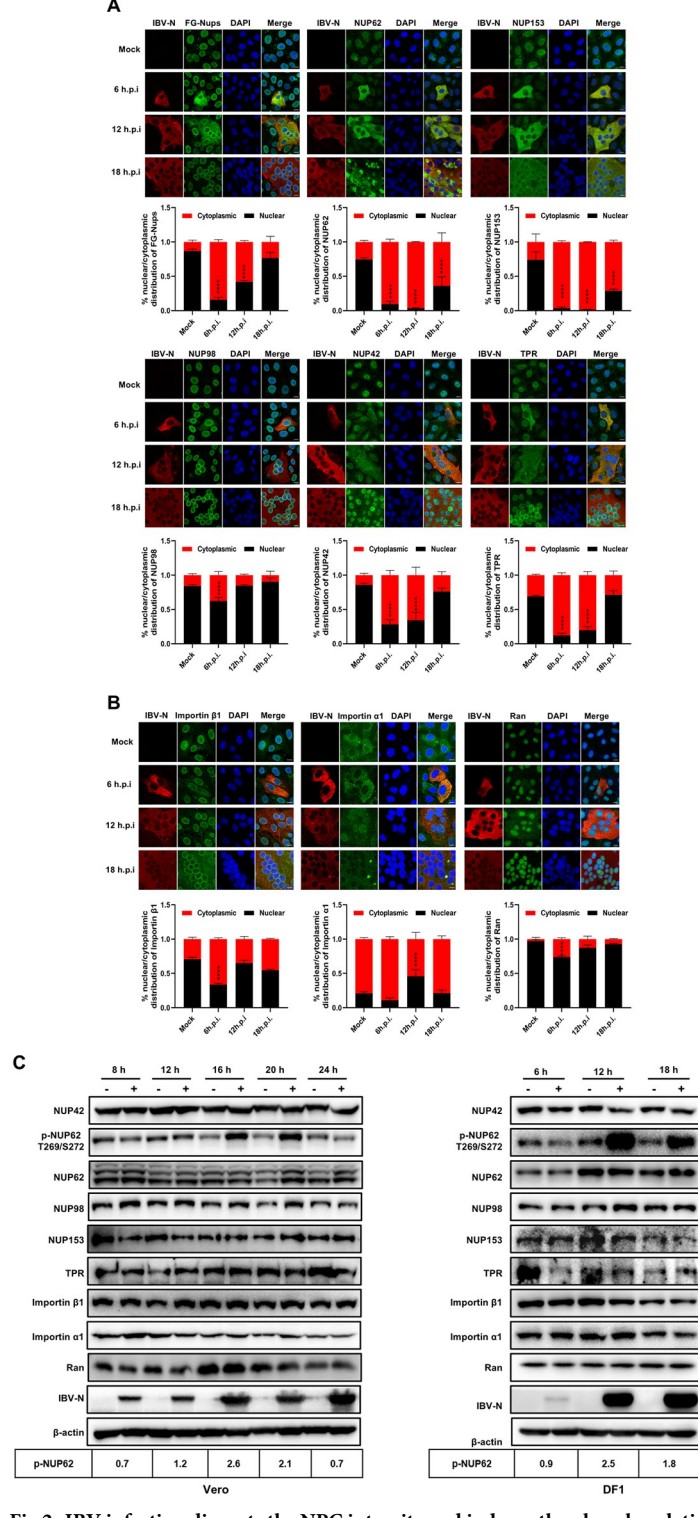

**Fig 2. IBV infection disrupts the NPC integrity and induces the phosphorylation of NUP62.** (A-B) Vero cells were infected with IBV or mock-infected, harvested at 6 h.p.i., 12 h.p.i., and 18 h.p.i., and subjected to immunofluorescence analysis. Representative images from three independent experiments are shown, with scale bars indicating 10 μm. The fluorescence signals of corresponding proteins (Nups, importins, and Ran) in mock-infected cells and IBV-infected cells from three fields of views were quantified using ImageJ. The intensities of the fluorescence signals in the nucleus (black bars) and the cytoplasm (red bars) are presented as bar graphs. Error bars represent the SD. Statistical

significance levels are denoted as follows: ****P < 0.0001. (C) Vero and DF-1 cells were infected with IBV or mock-infected, harvested at the indicated time points, and subjected to western blot analysis. β-actin was detected as a loading control. Protein band signals were quantified using ImageJ, with the intensities of p-NUP62 normalized to total NUP62. The ratio of p-NUP62 in IBV-infected cells to mock-infected cells is presented as p-NUP62 (+:-).

Next, we examined the expression levels of NUP42, NUP62, NUP98, NUP153, TPR, importin β1, importin α1, Ran, and phosphorylation level of NUP62 (p-NUP62), by western blot analysis. As depicted in Fig 2C, the expression levels of NUP42, NUP62, NUP98, NUP153, importin β1, importin α1, and Ran remained stable in both IBV-infected Vero cells and DF-1 cells. However, there was a mobility shift observed in NUP42, with the appearance of a smaller band, and phosphorylation of NUP62 at T269 and S272 was observed. These results suggest that IBV infection alters the post-translational state of NUP42 and NUP62. Notably, the phosphorylation level of NUP62 (Fig 2C) and the cytoplasmic diffusion of NUP98 (Figs 2A and S2) were more pronounced in DF-1 cells compared to Vero cells, suggesting that the disruption of the nucleocytoplasmic transport system caused by IBV infection is more evident in DF-1 cells. In summary, these findings suggest that IBV infection primarily disrupts nucleocytoplasmic trafficking function by inducing alterations in the post-translational modifications of Nups and their translocation from the nuclear envelope to the cytoplasm. The redistribution multiple Nups and importins to the cytoplasm likely contributes to the restriction of transcription factor nuclear entry during IBV infection process.

## The IBV N protein is implicated in disrupting nucleocytoplasmic trafficking and hindering the nuclear translocation of transcription factors

To identify viral proteins responsible for the cytoplasmic dispersion of Nups and the obstruction of transcription factor nuclear translocation, IBV proteins were tagged with Flag and expressed in Vero cells. The schematic depiction of IBV-encoded proteins is illustrated in Fig 3A. Immunofluorescence analysis revealed that overexpression of nsp2, nsp5, nsp8, nsp12, nsp13, nsp14, nsp15 and nsp16 did not impact the nuclear envelope localization of FG-Nups; conversely, in cells expressing nsp3, nsp6, nsp7, nsp9, E, M, 5a, and N, mislocalization of FG-Nups was observed (Fig 3B, left panel). Subsequently, we investigated whether these viral proteins hinder nuclear translocation of STAT1. As depicted in Fig 3B (right panel), nsp2, nsp5, nsp6, nsp12, nsp13, nsp15, E, M, 5a, and N notably reduced nuclear signals of STAT1 induced by IFNβ. Based on these findings, we conclude that nsp6, E, M, 5a and N not only facilitate the cytoplasmic dispersion of FG-Nups but also impede IFNβ-induced STAT1 nuclear translocation. Notably, among these viral proteins, the N protein prominently promotes the cytoplasmic distribution of FG-Nups, consistent with the observation in IBV-infected cells. Therefore, the IBV N protein emerges as a promising candidate for further investigation. It is noteworthy to mention that the expression of nsp4, nsp10, nsp11, S, 3a, 3b, and 5b was not successful in this study. Therefore, we cannot exclude the possibility that these viral proteins may also contribute to the perturbation of FG-Nups or the inhibition of STAT1 nuclear entry.

To further investigate the impact of IBV N protein on the subcellular localization of Nups, Vero cells were transfected with a plasmid encoding IBV N. The subcellular distributions of specific Nups were assessed with corresponding antibodies. Immunofluorescence analysis revealed that in cells expressing the N protein, the subcellular distributions of NUP42, NUP62, and NUP153 were dispersed from the nuclear envelope to the cytoplasm, showing significant colocalization with the N protein. Additionally, the signals of NUP98 and TPR were also

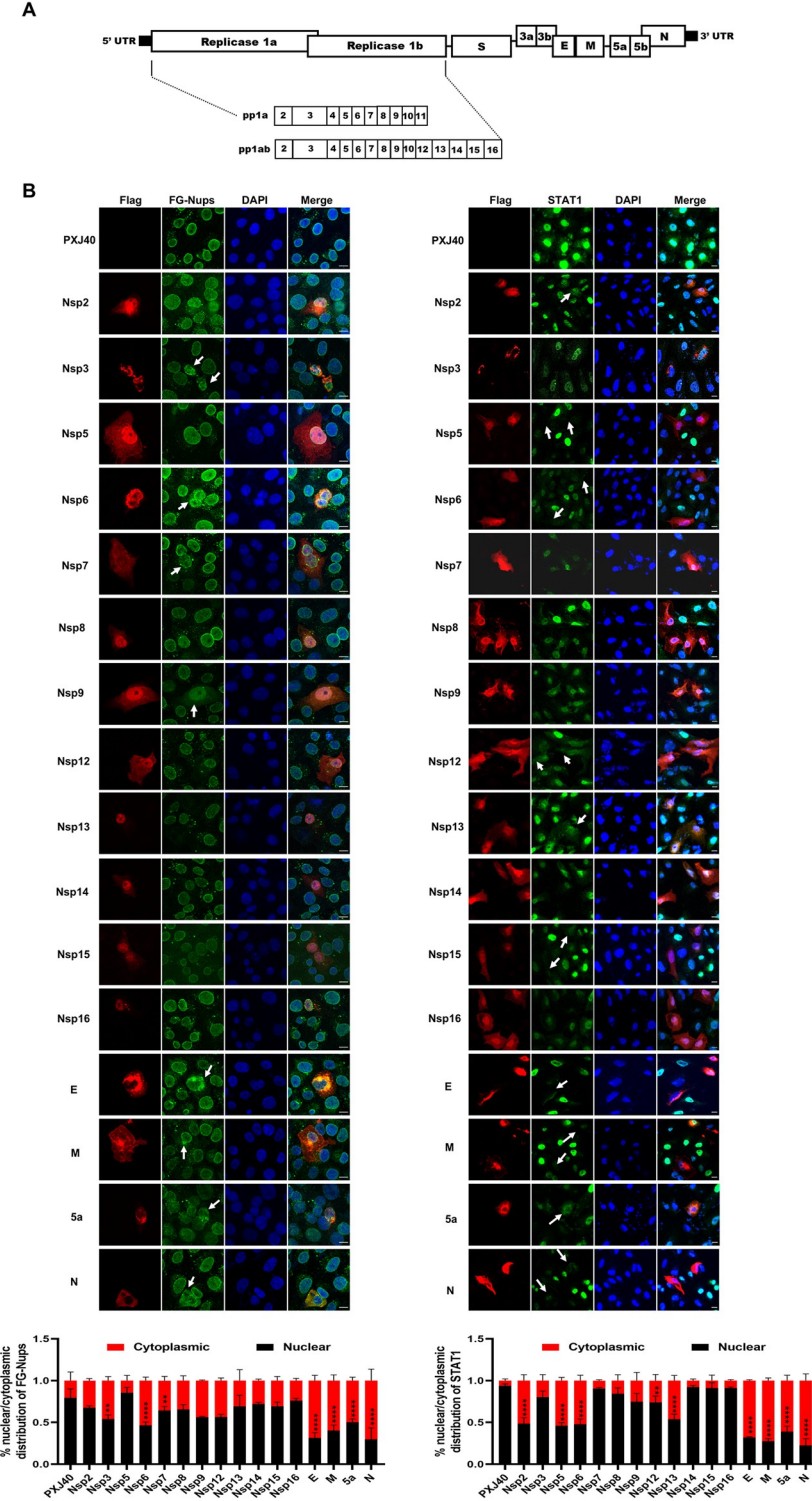

**Fig 3. Screening of IBV proteins that induce mislocalization of FG-Nups and inhibit IFNβ-induced nuclear translocation of STAT1.** (A) Schematic diagram of proteins encoded by IBV. (B) Vero cells were transfected with plasmids encoding Flag-tagged IBV proteins or vector PXJ40. At 24 h post-transfection, cells were subjected to immunofluorescence staining for the detection of FG-Nups using mAb414 (left panel). In a parallel group, transfected cells were treated with IFNβ (1000 IU/mL) for 40 min before immunofluorescence staining for detection of STAT1 (right panel). Representative images from three independent experiments are shown, with scale bars indicating 10 μm.

The fluorescence signals of FG-Nups and STAT1 in vector-transfected cells and IBV protein expressing cells from three fields of view were quantified using ImageJ. The intensities of the fluorescence signals in the nucleus (black bars) and the cytoplasm (red bars) are presented as bar graphs. Error bars represent the SD. Statistical significance levels are denoted as follows: ns, $P > 0.05$; *$P < 0.05$; **$P < 0.01$; ***$P < 0.001$; ****$P < 0.0001$.

altered: NUP98 lost its characteristic intact ring signal and appeared as punctate signals in the cytoplasm, while TPR was dispersed into the cytoplasm with reduced signal intensity (Fig 4A). These findings further support the notion that IBV N is the viral protein responsible for the cytoplasmic dispersion or mislocalization of Nups. Furthermore, importin β1, which is typically localized adjacent to the nuclear envelope as ring signals, exhibited altered morphology associated with the distorted nuclear envelope in cells expressing the N protein (Figs 4B and S3). Conversely, importin α1, which is primarily dispersed in the cytoplasm, exhibited reduced signals in cells expressing the N protein (Figs 4B and S3). Regarding Ran, whose signals are predominantly nuclear, they were relocated from the nucleus to the cytoplasm with diminished signal intensity in cells expressing the N protein (Fig 4B). Collectively, these findings provide evidence that the NPC integrity and nucleocytoplasmic trafficking function are perturbed by IBV N protein.

To ascertain whether N protein impedes the nuclear translocation of transcription factors, similar to those observations during IBV infection, Vero cells were transfected with the plasmid encoding Flag-tagged IBV N followed by poly(I:C) transfection. According to Fig 4C, the expression of IBV N protein significantly inhibited the poly(I:C)-induced nuclear translocation of IRF3, resulting in the retention of IRF3 in the cytoplasm. Further examination of the JAK-STAT pathway revealed that although IFNβ effectively stimulated the nuclear entry of STAT1, STAT2, and IRF9 in Vero cells, the expression N protein sequestered these transcription factors in the cytoplasm (Fig 4C). Evaluation of NF-κB pathway demonstrated that N protein expression impeded the TNFα-induced nuclear translocation of p65. Similarly, a blockage of UV irradiation-triggered nuclear translocation of p38 MAPK was observed in cells expressing N protein (Fig 4C).

Consequently, the transcription of IFNβ, IFITM3, and IL-8, induced by their respective stimuli, was significantly inhibited by the N protein in DF-1 cells (Fig 4D). These results collectively indicate that IBV N protein disrupts NPC integrity, thereby preventing the nuclear translocation of transcription factors and impairing the expression of downstream antiviral and pro-inflammatory genes. Additionally, we tested other IBV-encoded proteins for their inhibitory effects on antiviral gene expression. The results revealed that, besides the N protein, other viral proteins, particularly nsp5, nsp15, and 5a, also significantly inhibited both the interferon signaling pathway and the inflammatory response signaling pathway (S4 Fig). This indicates that the virus employs multiple strategies to counteract the host's innate immune response.

## The disruption of NPC integrity and interference with nuclear translocation of transcription factors by the N protein is conserved across diverse coronaviruses

The coronavirus N protein is a highly conserved structural protein responsible for encapsulating the viral genomic RNA and facilitating viral RNA synthesis/translation [57]. We aimed to investigate whether the disruption of nucleocytoplasmic trafficking by N protein is conserved across diverse coronaviruses. Flag-tagged N proteins from various genera of coronaviruses, including *alpha*-coronavirus HCoV-H229E, HCoV-NL63, TGEV and PEDV, *beta*-

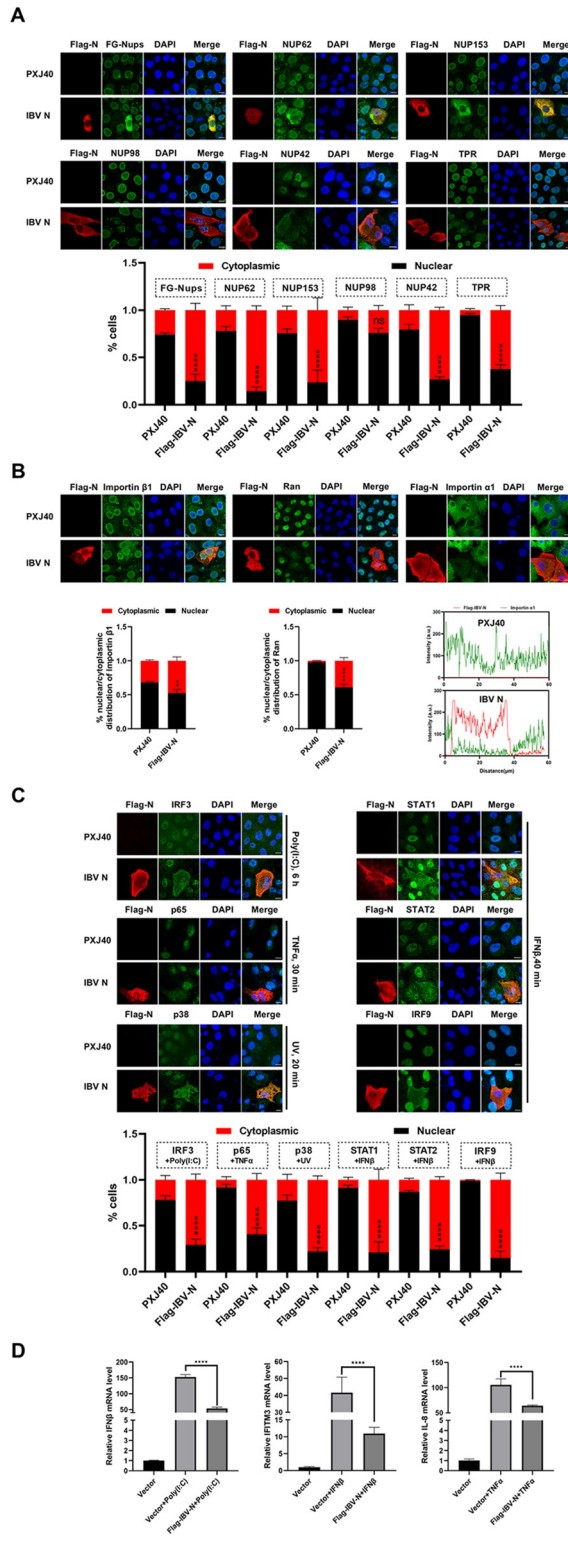

**Fig 4. IBV N protein induces dislocation of Nups from the nuclear envelope to the cytoplasm, inhibits nuclear translocation of transcription factors, and suppresses the transcription of antiviral genes.** (A-B) Vero cells were transfected with either the vector PXJ40 or a plasmid encoding IBV N protein. At 24 h post-transfection, cells were harvested and subjected to immunofluorescence analysis. (C) Vero cells were transfected with either vector PXJ40 or a plasmid encoding IBV N protein. At 18 h post-transfection, cells were further treated with poly(I:C) for 6 h. In parallel

experimental groups, cells were transfected with the vector PXJ40 or a plasmid encoding IBV N protein for 24 h, followed by treatment with IFNβ, TNFα, or UV irradiation, respectively. Cells were then harvested and subjected to immunofluorescence analysis. (A-C) Representative images from three independent experiments are shown, with scale bars indicating 10 μm. The fluorescence signals of corresponding proteins (Nups, importin β1, Ran and transcription factors) in vector-transfected cells and N expressing cells from three fields of view were quantified using ImageJ. The intensities of the fluorescence signals in the nucleus (black bars) and the cytoplasm (red bars) are presented as bar graphs. Error bars represent the SD. The relative fluorescence intensities of the IBV N protein and importin α1 were quantified using ImageJ. The quantitative data on the relative signal intensities and distributions of these two proteins are presented in the graph shown in the bottom panel of (B). Statistical significance levels are denoted as follows: ns, $P > 0.05$; *$P < 0.05$; **$P < 0.01$; ***$P < 0.001$; ****$P < 0.0001$. (D) DF-1 cells were transfected with the vector PXJ40 or a plasmid encoding IBV N protein. At 24 h post-transfection, cells were transfected with poly(I:C) or treated with IFNβ or TNFα for 12 h, followed by qRT-PCR analysis. Error bars represent the SD of technical triplicates. Statistical significance: ****$P < 0.0001$.

coronavirus MERS-CoV, SARS-CoV, MHV, PHEV, and SARS-CoV-2, and *delta*-coronavirus PDCoV, were expressed in Vero cells, and the subcellular localization of several Nups were examined. As shown in Fig 5A–5B, N proteins from various coronaviruses induced cytoplasmic dispersion of FG-Nups, NUP62, NUP153, and NUP42, with significant colocalization with the N protein. TPR exhibited a cytoplasmic dispersion pattern with reduced signal intensity in all N protein-expressing cells. Conversely, NUP98 lost its intact ring signal and formed punctate aggregates in the cytoplasm in the majority of N protein-expressing cells (including those from HCoV-NL63, MERS-CoV, SARS-CoV, MHV, PHEV, SARS-CoV-2, and PDCoV). The subcellular localization of two importins and Ran was also assessed. As illustrated in Fig 5C, in all N protein-expressing cells, importin β1 lost its intact ring signals on the nuclear envelope and displayed as mislocated signal. Additionally, a decreased signal of importin α1 was observed, and Ran was dispersed into the cytoplasm with diminished signal. Taken together, these data suggest that the ability to dismantle the NPC and interfere with nucleocytoplasmic trafficking is conserved across diverse coronaviruses N proteins.

Subsequently, we investigated whether the nuclear translocation of transcription factors is impeded by N proteins from various coronaviruses. Vero cells were transfected with N proteins, followed by poly (I:C) transfection. As shown in Fig 5D, poly(I:C) successfully induced the nuclear translocation of IRF3; however, in all N protein-expressing cells, IRF3 remained in the cytoplasm. Further examination of the JAK-STAT pathway also revealed that, although IFNβ successfully stimulated the nuclear translocation of STAT1, STAT2, and IRF9, in all N protein-expressing cells, STAT1, STAT2, and IRF9 were dispersed in the cytoplasm (Fig 5E). Moreover, p65 and p38 MAPK entered the nucleus following TNFα treatment or UV irradiation; however, the expression of N proteins impeded the nuclear translocation of p65 and p38 MAPK (Fig 5D). The NLS for IRF3 is RTQKRLR, for STAT1 is KGKKTK, for STAT2 is KRKRK, for IRF9 is KRKR, and for p65 is RKRR. The inhibition of nuclear import of tandem GFP-GFP directed by the SV40 large T antigen NLS sequence, PKKKRKV, in N protein-expressing cells further demonstrated that the N protein inhibits the nuclear import of proteins containing various NLS sequences (Fig 5F). Alongside these observations, the transcription of IFNβ, IFITM3 and IL-8 in response to corresponding stimuli were significantly inhibited by all N proteins from different coronaviruses (Fig 5G). Overall, these findings demonstrate the blockage of multiple transcription factors' nuclear translocation and inhibition of antiviral gene expression by the N protein is conserved across diverse coronaviruses.

We further examined the effect of the N protein on mRNA export and protein translation. Vero cells were transfected with either vector control or plasmids encoding the N proteins from various coronaviruses. The cells were then subjected to two sets of analyses: (1)

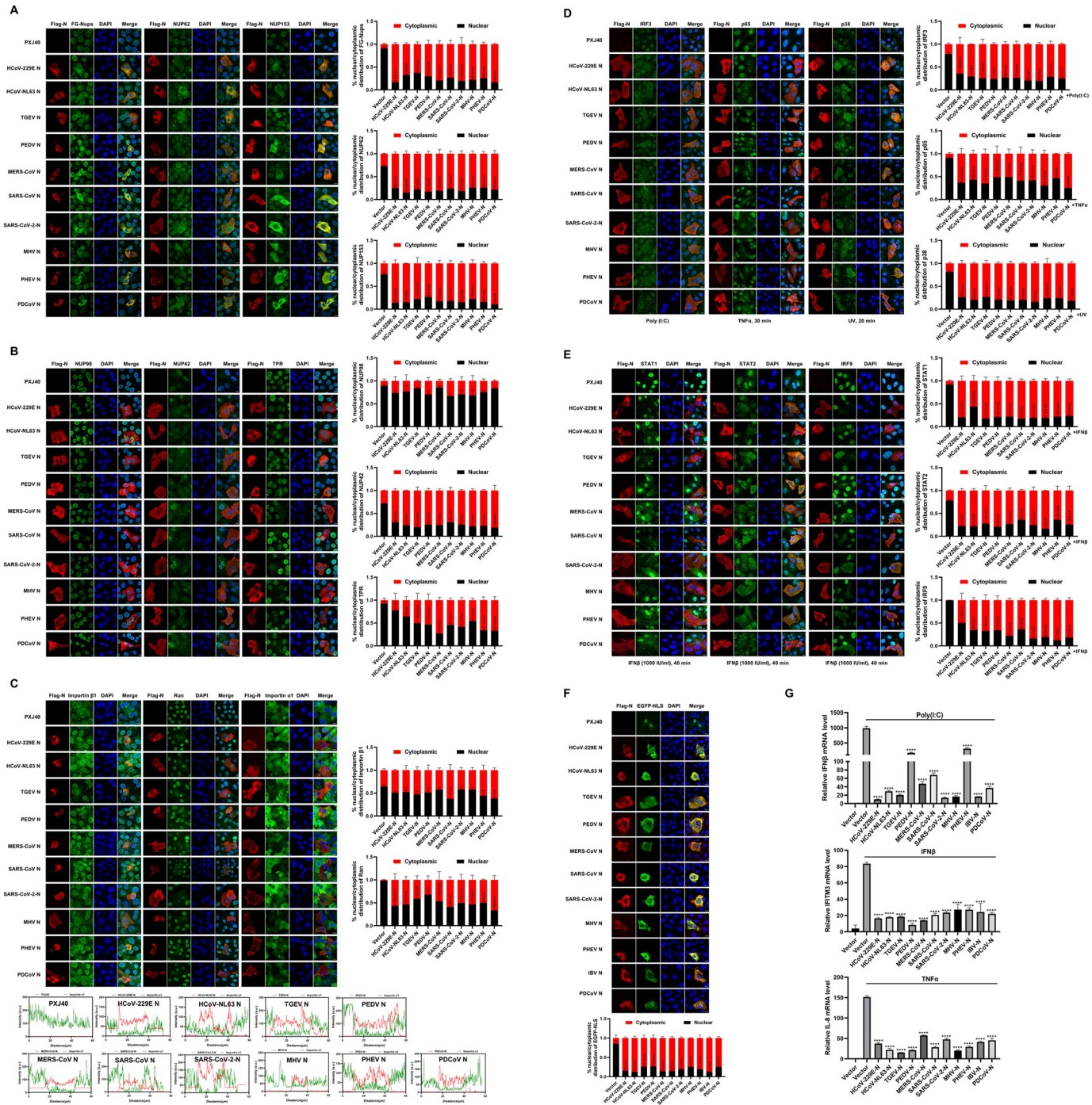

**Fig 5. Disruption of nucleocytoplasmic trafficking, inhibition of nuclear translocation of transcription factors, and suppression of antiviral genes transcription by diverse coronaviruses N proteins.** (A-C) Vero cells were transfected with plasmids encoding Flag-tagged N proteins from different genera of coronaviruses, while vector PXJ40 served as a control. Cells were harvested 24 h post-transfection and subjected to immunofluorescence analysis. (D-E) Vero cells were transfected with plasmids encoding N proteins or vector PXJ40. At specific time points after transfection, cells were treated with poly(I:C), IFNβ, TNFα, or UV irradiation, respectively, followed by immunofluorescence analysis. (F) Vero cells were transfected with plasmids encoding N proteins or vector PXJ40, together with plasmid encoding tandem GFP-GFP. Cells were harvested 24 h post-transfection and subjected to immunofluorescence analysis. (A-F) Representative images from three independent experiments are shown, with scale bars indicating 10 μm. The fluorescence signals of corresponding proteins in vector-transfected cells and N expressing cells from three fields of view were quantified using ImageJ. The intensities of the fluorescence signals in the nucleus (black bars) and the cytoplasm (red bars) are presented as bar graphs. Error bars represent the SD. The relative fluorescence intensities of the N protein and importin α1 were quantified using ImageJ. The quantitative data on the relative signal intensities and distributions of these two proteins are presented in the graph shown in the bottom panel of (C). Statistical significance levels are denoted as follows: ns, P > 0.05; *P < 0.05; **P < 0.01; ***P < 0.001; ****P < 0.0001. (G) HEK-

293T cells were transfected with plasmids encoding N proteins or vector PXJ40. At 24 h post-transfection, cells were treated with poly(I:C), IFNβ, or TNFα. Cells were harvested at 12 h post-treatment, and the transcription levels of IFNβ, IFITM3, or IL-8 were detected using qRT-PCR analysis.

fluorescence in situ hybridization (FISH) and immunofluorescence to assess the subcellular localization of mRNA, and (2) puromycin labeling and immunofluorescence to monitor real-time protein translation. As shown in S5A Fig, N proteins from HCoV-229E, HCoV-NL63, TGEV, PEDV, MHV, IBV, and PDCoV effectively inhibited the nuclear export of mRNA. In contrast, N proteins from MERS-CoV, SARS-CoV, SARS-CoV-2, and PHEV had a less pronounced effect on mRNA nuclear retention. Correspondingly, a decrease in real-time protein synthesis, as indicated by puromycin labeling, was observed in cells expressing N proteins from HCoV-229E, HCoV-NL63, TGEV, PEDV, MERS-CoV, MHV, PHEV, IBV, and PDCoV (S5B Fig). Notably, the N proteins from SARS-CoV, SARS-CoV-2, and MHV, did not significantly inhibit protein translation. These results suggest that the N protein regulates the nuclear transport system by not only inhibiting the nuclear import of transcription factors but also affecting mRNA export and subsequent protein synthesis. However, there are slight differences in the efficacy of various N proteins in performing these functions, particularly with the N proteins from SARS-CoV and SARS-CoV-2, which displayed a reduced capacity to inhibit protein translation.

## Diverse coronaviruses N proteins interact with scaffold protein RACK1

To delve into the mechanisms underlying how the N protein modulates the nucleocytoplasmic trafficking, we conducted co-immunoprecipitation (Co-IP) assays combined with liquid chromatography and mass spectrometry (LC-MS/MS) to screen cellular proteins interacting with IBV N protein. A comprehensive analysis identified a total of 694 cellular candidates that co-immunoprecipitated with IBV N protein. Gene ontology (GO) annotation and Kyoto Encyclopedia of Genes and Genomes (KEGG) database analyses revealed a significant enrichment of host gene expression pathways. These include mitochondrial translation, mRNA metabolic process, ribosome biogenesis, regulation of translation, ribonucleoprotein complex assembly, RNA localization, spliceosomal complex assembly, RNA modification, RNA 3'-end processing, among others (Fig 6A-B). Notably, these findings align with the previous interactome analyses of the N proteins from IBV [58], PEDV [59], SARS-CoV-2 [60, 61]. Among the IBV N binding proteins involved in nucleocytoplasmic transport, key players were identified and listed in Fig 6C, including RACK1 (Receptor of activated protein C kinase 1), NXF1 (Nuclear RNA export factor 1), LMNB1 (Lamin B1), KPNA2 (importin α1), PPP1CC (Protein phosphatase PP1γ), RAE1 (mRNA export factor), LBR (Lamin B receptor), SRPK1 (SRSF protein kinase 1), NUPL2 (Nucleoporin-like protein 2, NUP42).

Among the proteins interacting with IBV N, as listed in Fig 6C, RACK1 stands out as highly conserved intracellular adaptor protein with pivotal roles in anchoring and stabilizing proteins activity, and shuttling proteins to specific cellular location, such as activated PKC [62]. To validate the interaction between N protein and RACK1, we co-transfected plasmids encoding Flag-tagged IBV N and HA-tagged RACK1 into HEK-293T cells. The specific binding between IBV N protein and RACK1 was confirmed by Co-IP using anti-HA or anti-Flag antibodies: both antibodies successfully precipitated Flag-N and HA-RACK1 together (Fig 7A). We further examined whether IBV N binds to endogenous RACK1 during virus infection. DF-1 cells were infected with IBV, followed with immunoprecipitation using anti-IBV N polyclonal antibody. The results demonstrated that the anti-IBV N antibody efficiently co-precipitated N protein and endogenous RACK1, providing evidence of specific binding during the virus

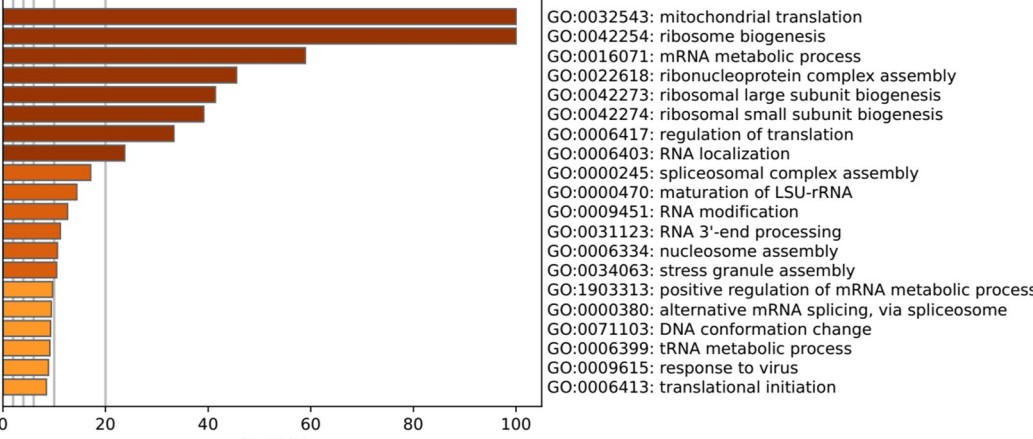

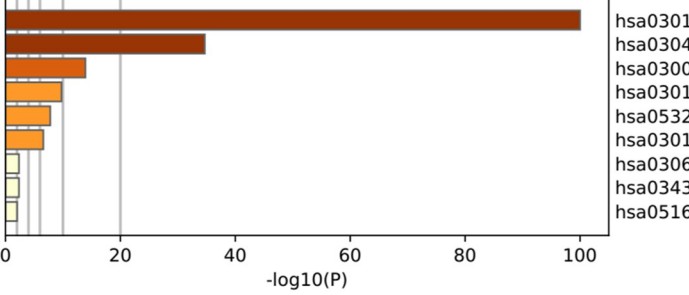

**Fig 6. Bioinformatic analysis of IBV N-protein interactome.** Plasmid encoding IBV N protein was transfected into HEK-293T cells for 30 h. Whole-cell lysates were immunoprecipitated using anti-Flag antibody, and three independent immunoprecipitated samples were subjected to LC-MS/MS analysis. PXJ40-transfected cell lysates served as negative control to eliminate nonspecific binding proteins. (A) Heatmap illustrating the enrichment of Gene Ontology (GO) annotations within the IBV N-protein interactome. (B) Heatmap displaying the enrichment of Kyoto Encyclopedia of Genes and Genomes (KEGG) pathways in the IBV N-protein interactome. (C) IBV N protein interacts with several cellular proteins involved in nucleocytoplasmic transport.

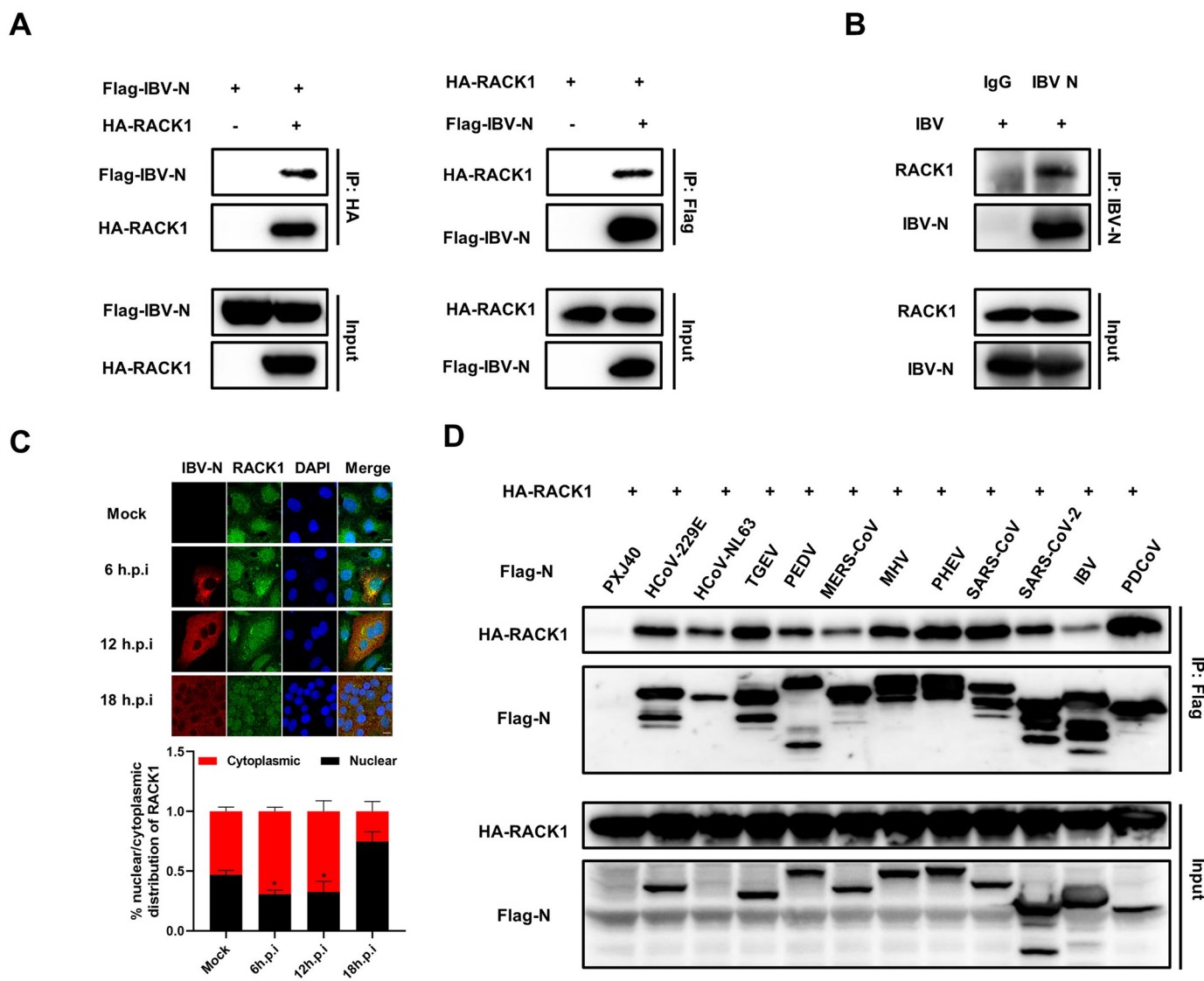

**Fig 7. Diverse coronaviruses N proteins interact with RACK1.** (A) HEK-293T cells were co-transfected with plasmids encoding Flag-IBV N and HA-RACK1. Co-transfections with Flag-IBV-N and PCMV-HA, or PXJ40 and HA-RACK1 were performed as control groups. Cell lysates were immunoprecipitated using anti-Flag or anti-HA antibodies, followed by immunoblot analysis. (B) DF-1 cells were infected with IBV at an MOI of 1. At 12 h.p.i., cell lysates were immunoprecipitated using anti-IBV N antibody, followed by immunoblot analysis. (C) Vero cells were infected with IBV at an MOI of 1 or mock-infected. Immunostaining was performed at 6 h.p.i., 12 h.p.i., and 18 h.p.i. Representative images from three independent experiments are shown. Scale bars: 10 μm. The fluorescence signals of RACK1 in mock-infected cells and IBV-infected cells from three fields of view were quantified using ImageJ. The intensities of the RACK1 fluorescence signals in the nucleus (black bars) and the cytoplasm (red bars) are presented as bar graphs. Error bars represent the SD. Statistical significance levels are denoted as follows: ns, P > 0.05; *P < 0.05. (D) HEK-293T cells were co-transfected with plasmids encoding Flag-tagged N proteins from different coronaviruses and HA-RACK1. Co-transfection of PXJ40 with plasmid encoding HA-RACK1 served as a control. HEK-293T cell lysates were immunoprecipitated using anti-Flag antibody, followed by immunoblot analysis.

infection process (Fig 7B). Since the antibody against human RACK1 recognizes the linear epitopes of chicken RACK1 but not its conformational epitopes, DF-1 cells are not suitable for studying the subcellular localization of endogenous RACK1. Therefore, Vero cells were employed for subsequent immunofluorescence study. The results in Fig 7C revealed that in mock-infected Vero cells, endogenous RACK1 was distributed in both the nucleus and cytoplasm. However, in IBV-infected Vero cells, a proportion of endogenous RACK1 was

dispersed into the cytoplasm at 6 h.p.i. and 12 h.p.i., where it co-localized with IBV N protein, indicating alterations in its localization by N protein during infection.

To investigate whether the interaction with RACK1 is a common feature across diverse coronaviruses N proteins, Flag-tagged N proteins from various coronaviruses were co-expressed with HA-RACK1 in HEK-293T cells. Co-IP results showed that the anti-Flag antibody efficiently precipitated both Flag-N proteins and HA-RACK1, while no RACK1 was pulled down in the absence of N protein (Fig 7D), indicating that the interaction with RACK1 is a conserved characteristic across diverse coronaviruses N proteins.

## The cytoplasmic redistribution of Nups requires PKCα/β activity

RACK1 serves as an anchoring protein for PKC and is responsible for trafficking PKC to specific subcellular locations. Although PKC has been reported to be involved in the modulation of lamin B1 phosphorylation and the cell cycle [63–65], whether PKC phosphorylates Nups remains unexplored. Notably, phosphorylation and cytoplasmic distribution of NUP62 were observed in IBV-infected cells, and N protein was found to be responsible for the cytoplasmic distribution of Nups. Intriguingly, N protein interacts with the PKC scaffold protein RACK1. These observations prompt us to speculate that N protein might regulate PKC activity via RACK1 and be involved in NUP62 phosphorylation. To test our hypothesis, we first examined whether PKC activity is implicated in the cytoplasmic distribution of Nups induced by N protein expression. Vero cells were transfected with Flag-tagged N proteins from different coronaviruses and the subcellular localization of PKCα/β, FG-Nups, NUP62, and NUP153 was examined. Immunofluorescence analysis revealed that PKCα/β translocated from the nucleus to the cytoplasm together with FG-Nups, NUP62, or NUP153 in all N protein-expressing cells (Fig 8A–8C), demonstrating that N protein alters the subcellular localization of PKCα/β and affects their activity. The pronounced colocalization of N, PKCα/β, and FG-Nups/NUP62/NUP153 suggests potential interactions among N, kinases and Nups. The application of the PKCα/β inhibitor Enzastaurin resulted in the majority of PKCα/β, FG-Nups, NUP62, and NUP153 relocating to the nucleus or perinuclear region, while all N proteins remained diffusely distributed throughout the cytoplasm (Fig 8A–8C). Thus, the activity of PKCα/β is implicated in the cytoplasmic dispersion of Nups in N protein-expressing cells.

To further confirm the implication of PKC in the cytoplasmic distribution of Nups, HA-tagged PKCα and PKCβ were expressed in Vero cells alone or together with IBV N protein. Immunofluorescence analysis revealed that overexpression of both HA-PKCα and HA-PKCβ promoted the cytoplasmic distribution of NUP62, while the cytoplasmic dispersion of FG-Nups and NUP153 was less pronounced. Surprisingly, co-expression of IBV N protein and PKCα/PKCβ led to enhanced cytoplasmic distribution of FG-Nups, NUP62, and NUP153, with these Nups colocalizing well with N and PKCα/PKCβ (Fig 8D). This observation suggests that PKCα and PKCβ are capable of inducing the cytoplasmic dispersion of NUP62, and the presence of N protein promotes the dispersion of more Nups into the cytoplasm.

## The activity of PKCα/β is implicated in the phosphorylation of NUP62 and the cytoplasmic distribution of Nups during IBV infection

To determine whether IBV infection activates PKC activity, we evaluated the phosphorylation levels of PKCα and PKCβ in IBV-infected Vero cells and DF-1 cells. As illustrated in Fig 9A, compared to mock-infected cells, the phosphorylation at S660 of PKCβ gradually increased as the infection progressed, peaking at 18 h.p.i.. In contrast, the phosphorylation level at T638 of PKCα remained stable throughout the infection. Notably, the level of p-NUP62 (at T269/S272) also exhibited a gradual increase during IBV infection, concurrent with the phosphorylation of

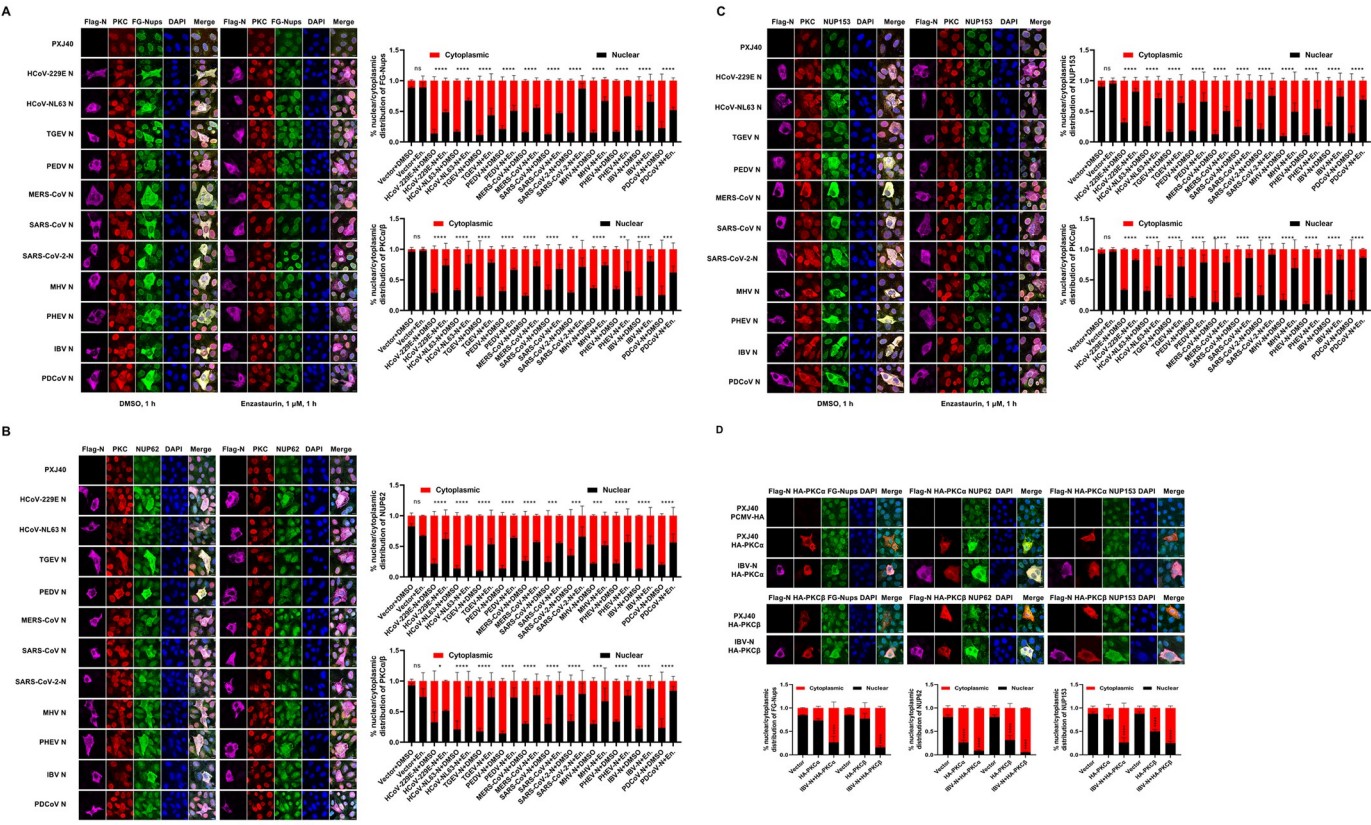

**Fig 8. Implication of PKCα/β activity in the cytoplasmic distribution of Nups.** (A-C) Vero cells were transfected with plasmids encoding Flag-tagged N proteins from different coronaviruses or PXJ40. After 24 h post-transfection, cells were treated with Enzastaurin (1 μM) or DMSO for 1 h, followed by immunostaining to assess the subcellular distribution of N, PKCα/β, and FG-Nups/NUP62/NUP153. (D) Plasmid encoding HA-tagged PKCα or PKCβ was co-transfected with PXJ40, or co-transfected with plasmid encoding Flag-tagged IBV N for 24 h, followed by immunostaining to evaluate the subcellular localization distribution of PKCα, PKCβ, N, FG-Nups, NUP62, and NUP153. Representative images from three independent experiments are shown. Scale bars: 10 μm. The fluorescence signals of Nups and PKC in vector-transfected cells and N expressing cells from three fields of view were quantified using ImageJ. The intensities of the fluorescence signals in the nucleus (black bars) and the cytoplasm (red bars) are presented as bar graphs. Error bars represent the SD. Statistical significance levels are denoted as follows: ns, P > 0.05; ****P < 0.0001.

PKCβ. It was observed that the phosphorylation levels of PKCβ and NUP62 are more pronounced in DF-1 cells compared to Vero cells. The mechanisms underlying these differences between the cell lines remain unclear. Immunofluorescence analysis, as shown in Fig 9B top panel, revealed that during the early stages of infection (6 h.p.i. and 12 h.p.i.), PKCα/β translocated from the nucleus to the cytoplasm and displayed diffuse colocalization with the N protein in Vero cells. However, as the infection progressed (from 12 h.p.i. to 18 h.p.i.), PKCα/β gradually relocated back to the nucleus, suggesting a dynamic regulation of their subcellular localization by IBV infection in Vero cells. This alteration in the subcellular localization of PKCα/β, induced by the virus, likely determines their proximity to substrates, thereby influencing their kinase activity. It was observed that in DF-1 cells (Fig 9B, bottom panel), there was no obvious nuclear localization of PKCα/β in mock-infected cells, suggesting that the subcellular localization of PKCα/β in chicken cells differs from that in Vero cells under physiological conditions. After IBV infection, PKCα/β exhibited diffuse colocalization with the IBV N protein in the cytoplasm at 6 h.p.i. and 12 h.p.i., indicating an interaction between PKCα/β and the N protein (Fig 9B, bottom panel). This phenomenon may explain the higher levels of

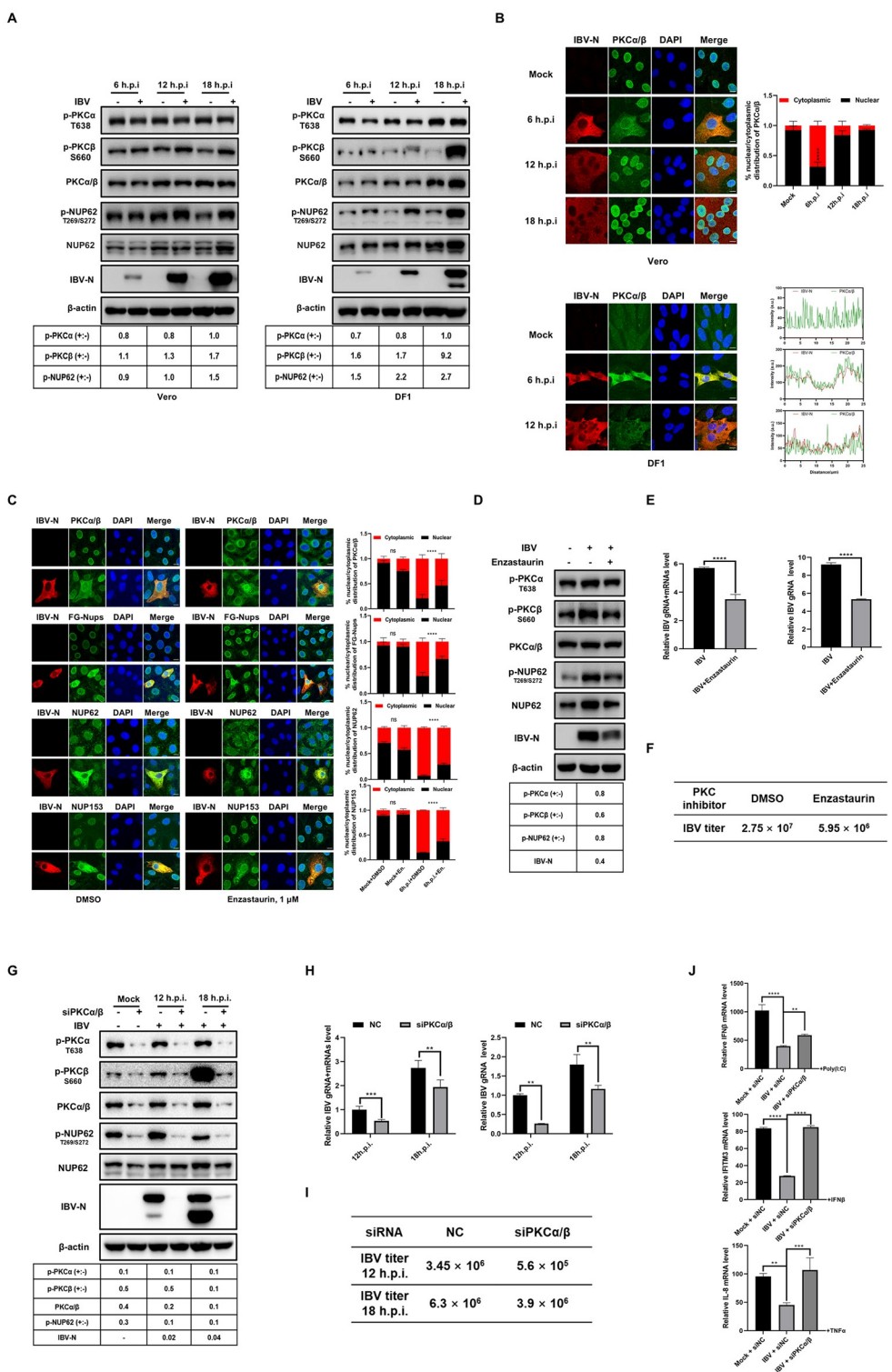

**Fig 9. Role of PKCα/β activity in NUP62 phosphorylation, the cytoplasmic distribution of FG-Nups, suppression of the antiviral response, and virus replication.** (A) Vero or DF-1 cells were infected with IBV at an MOI of 1 or mock-infected. Cells were harvested at 6, 12, and 18 h.p.i. and subjected to Western blot analysis using corresponding antibodies. The intensities of p-PKCα, p-PKCβ, and p-NUP62 bands were normalized to total PKCα/β or NUP62, respectively. The ratio of p-PKCα, p-PKCβ, and p-NUP62 in IBV-infected cells to those in mock-infected cells is denoted as p-PKCα (+:-), p-PKCβ (+:-), and p-NUP62 (+:-). (B) Vero and DF-1 cells were infected with IBV or mock-

infected and subjected to immunostaining at indicated time point. (C) Vero cells were infected with IBV or mock-infected. At 2 h.p.i., cells were treated with DMSO or 1μM Enzastaurin for 4 h. The subcellular localization of the indicated proteins was analyzed with immunostaining at 6 h.p.i.. (B-C) Representative images from three independent experiments are shown, with scale bars indicating 10 μm. The fluorescence signals of PKCα/β and Nups in mock-infected and IBV-infected Vero cells were quantified from three fields of view using ImageJ. The intensities of the fluorescence signals in the nucleus (black bars) and the cytoplasm (red bars) are presented as bar graphs, with error bars representing the SD. The fluorescence intensities of IBV N protein and PKCα/β in mock-infected and IBV-infected DF-1 cells were quantified using ImageJ. The relative distributions of these proteins are presented in the graph in the right panel of (B). (D-F) DF-1 cells were infected with IBV or mock-infected, followed by treatment with DMSO or 1 μM Enzastaurin at 2 h.p.i. At 18 h.p.i., Western blot analysis was performed using the indicated antibodies (D). RT-PCR was conducted with primers targeting IBV gene 1 and gene N to measure the levels of IBV genome and transcripts (E). Additionally, the culture medium was collected and subjected to a plaque assay to determine IBV titers (F). (G-I) DF-1 cells were transfected with siRNA targeting PKCα/β or non-target control siRNA (NC) for 48 h, followed by IBV infection at 12 h.p.i. and 18 h.p.i. Western blot analysis was performed to detect the indicated proteins (G). RT-PCR was conducted with primers specific to IBV gene 1 and gene N to measure the levels of IBV genome and transcripts (H). The culture medium was collected and analyzed using a plaque assay to determine IBV titers (I). The ratio of p-PKCα, p-PKCβ, p-NUP62, and N protein levels in Enzastaurin-treated or siPKCα/β-transfected cells compared to those in DMSO-treated or NC-transfected cells is denoted as p-PKCα (+:-), p-PKCβ (+:-), NUP62 (+:-), and N (+:-) (D and G). (J) DF-1 cells were transfected with siRNA targeting PKCα/β or siRNA (NC) for 48 h, followed by IBV infection at an MOI of 5. The infected cells were then treated with poly(I:C) or TNFα at 2 h.p.i. for 6 h, or with IFNβ at 6 h.p.i. for 6 h. qRT-PCR was conducted to measure the transcription levels of IFNβ, IFITM3, and IL-8. For panel E, H, and J, error bars represent the SD of technical triplicates. Statistical significance levels are denoted as follows: ns (not significant), P > 0.05; *P < 0.05; **P < 0.01; ***P < 0.001; ****P < 0.0001.

PKCβ phosphorylation observed in IBV-infected DF-1 cells compared to Vero cells, as shown in Fig 9A.

The next question we addressed was whether PKCα/β activity is implicated in the cytoplasmic dispersion of Nups during IBV infection. We treated IBV-infected Vero cells with the PKCα/β inhibitor Enzastaurin and examined the subcellular localization of Nups. As shown in Fig 9C, compared to the DMSO-treated group, Enzastaurin treatment caused PKCα/β, FG-Nups, NUP62, and NUP153 to relocate to the nucleus and perinuclear region in IBV-infected Vero cells, resulting in reduced cytoplasmic dispersion. These observations suggest that PKCα/β activity is necessary for IBV infection-induced Nups cytoplasmic redistribution, similar to what was observed in N protein-expressing cells. Interestingly, Enzastaurin treatment reduced the phosphorylation levels of PKCβ, NUP62, and the expression of the IBV N protein (Fig 9D). We further measured the relative levels of virus RNA genome and transcripts using RT-PCR. As shown in Fig 9E, Enzastaurin treatment decreased the copies of IBV RNA genome and transcripts compared to DMSO treatment. A plaque assay further demonstrated that the virus titer in the culture medium was reduced in Enzastaurin-treated cells (Fig 9F). Altogether, these results indicate that the activity of PKCα/β is involved in the phosphorylation of NUP62, the distribution of nucleoporins, and the support of viral replication.

The impact of PKCα/β on NUP62 phosphorylation was further investigated by suppressing PKCα and PKCβ expression using a siRNA targeting both kinases. As shown in Fig 9G, transfection with PKCα/β siRNA effectively suppressed the expression of both PKCα and PKCβ, resulting in a substantial reduction in total PKCα/β levels, as well as phosphorylated PKCα and PKCβ. Concurrently, the phosphorylation of NUP62 decreased to nearly undetectable levels in both mock- and IBV-infected cells, highlighting the essential role of PKCα/β in NUP62 phosphorylation under normal physiological conditions and during viral infection. Furthermore, the depletion of PKCα/β led to a significant decrease in IBV N protein levels (Fig 9G), viral genome and transcript levels (Fig 9H), and virus particle release (Fig 9I), underscoring the necessity of PKCα/β for efficient virus replication.

To investigate whether phosphorylation of NUP62 is responsible for its cytoplasmic distribution, we treated Vero cells with the PP1 and PP2A inhibitor okadaic acid to inhibit

dephosphorylation events. We examined the phosphorylation levels of PKCα, PKCβ, and NUP62 by Western blot analysis. As shown in S6A Fig, okadaic acid treatment led to a slight accumulation of p-PKCα, accompanied by the appearance of two additional bands corresponding to 110 kDa and 180 kDa, which represent dimers/trimers with enhanced activity [66, 67]. The phosphorylation form of PKCβ was also greatly accumulated. Therefore, PP1 or PP2A is implicated in the dephosphorylation of PKCα/PKCβ and suppression of their activity. Simultaneously, the phosphorylation form of NUP62 was predominantly accumulated. Meanwhile, a band with slower mobility at approximately 120 kDa was detected, which might represent a potential dimer. The dimerization or oligomerization of p-NUP62 is involved in self-interaction or interaction with other Nups [68, 69], although the underlying mechanism and functional consequence are unclear. Total NUP62 exhibited minor mobility shift bands representing hyperphosphorylation forms at multiple sites. The significant accumulation of p-NUP62 by okadaic acid treatment demonstrates that PP1 or PP2A directly dephosphorylates NUP62 and probably contribute to the assembly and disassembly of NPC. Immunofluorescence analysis showed that okadaic acid treatment greatly promoted the cytoplasmic distribution of NUP62 (S6B Fig, top panel), confirming the correlation between phosphorylation of NUP62 and its cytoplasmic dispersion. Additionally, a small proportion of FG-Nups was also dispersed to the cytosol in okadaic acid-treated cells (S6B Fig, bottom panel), further supporting the idea that phosphorylation events lead to the dissociation of FG-Nups.

Finally, we examined whether the depletion of PKCα/β affects the expression profile of antiviral genes. Real-time qRT-PCR results revealed that IBV infection compromised the expression of IFNβ, IFITM3, and IL-8 induced by their respective stimuli. However, in PKCα and PKCβ knockdown cells, the transcription levels of these antiviral genes were significantly restored (Fig 9J). This observation suggests that PKCα/β plays a role in the suppression of the innate immune response during IBV infection.

Overall, these results demonstrate that IBV infection activates PKCβ through phosphorylation and simultaneously alters the subcellular localization of both PKCα and PKCβ. The activated PKCα/β is then responsible for phosphorylating NUP62 and redistributing nucleoporins to the cytoplasm, which ultimately suppresses the expression of antiviral genes and facilitates virus replication. However, due to the non-specific knockdown or inhibition of PKCα and PKCβ by siRNA or chemical inhibitors, the current data do not allow us to determine whether PKCα or PKCβ specifically mediates the phosphorylation of NUP62.

## RACK1 is essential for the phosphorylation of PKCα/β and NUP62, the suppression of antiviral gene expression, and the promotion of IBV infection

To investigate whether RACK1 is involved in regulating PKC activity and NUP62 phosphorylation, we knocked down RACK1 in DF-1 cells, followed by IBV infection. As shown in Fig 10A, reduced RACK1 expression led to decreased levels of p-PKCα, p-PKCβ, and p-NUP62 in both mock- and IBV-infected DF-1 cells, demonstrating that RACK1 is involved in the phosphorylation of PKCα, PKCβ, and NUP62. Additionally, the expression of the IBV N protein (Fig 10A), viral genome and transcripts (Fig 10B), and released virus particles (Fig 10C) were all reduced in RACK1 knockdown cells, highlighting the importance of RACK1 in virus infection. Furthermore, depletion of RACK1 significantly restored the transcription of IFNβ, IFITM3, and IL-8 in IBV-infected DF-1 cells, regardless of chemical stimuli (Fig 10D–10E). These results indicate that RACK1 is a crucial host factor for maintaining PKCα and PKCβ kinase activity, which is necessary for phosphorylating NUP62, suppressing antiviral gene expression, and promoting IBV replication.

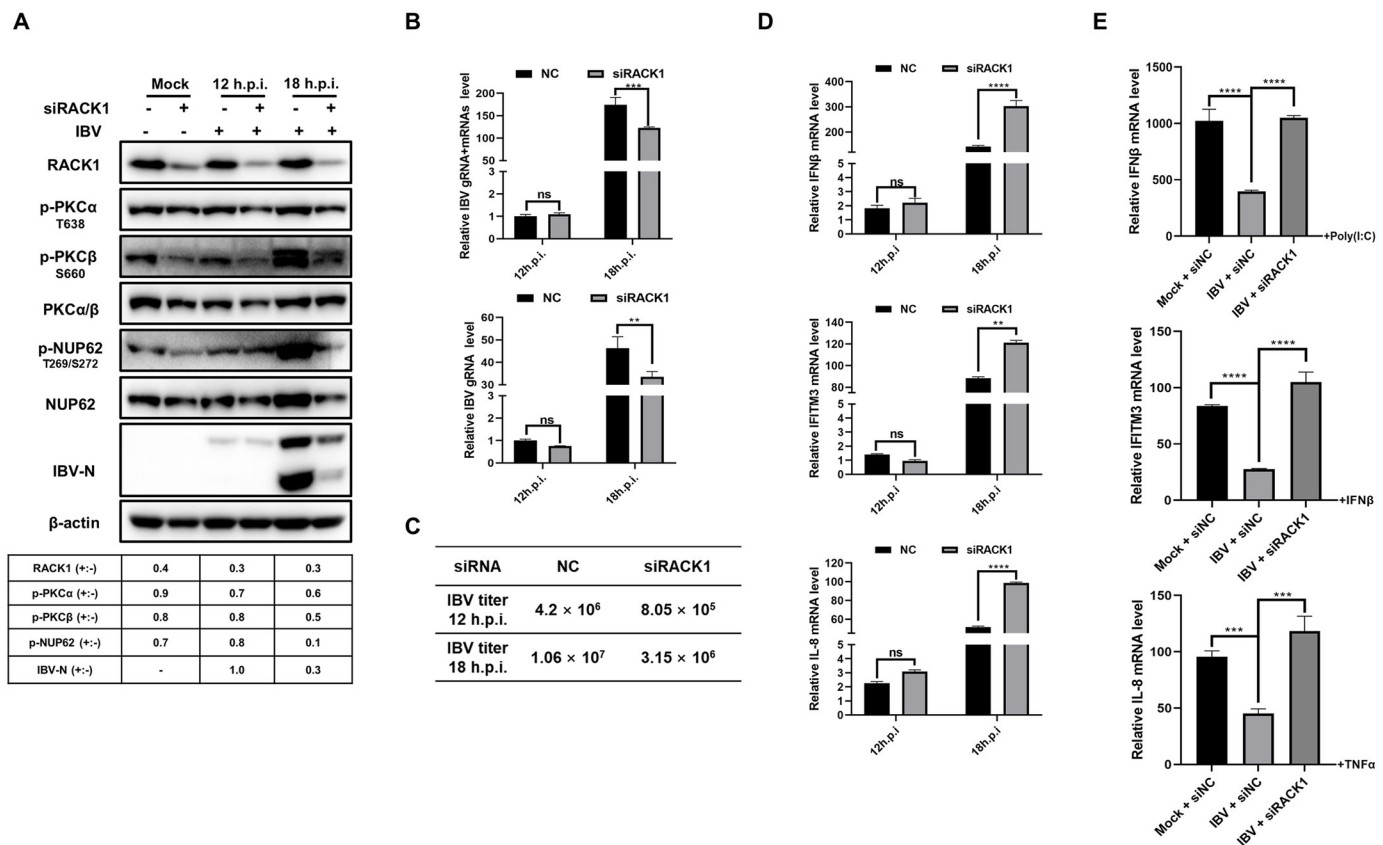

**Fig 10. Role of RACK1 in maintaining phosphorylation of PKCα, PKCβ, and NUP62, suppressing antiviral gene expression, and promoting IBV infection.**
(A-D) DF-1 cells were transfected with RACK1 siRNA or control siRNA (NC), followed by IBV infection or mock infection at 48 h post-transfection. Cells were harvested at the indicated time points. Western blot analysis was performed to examine the phosphorylation and expression of the indicated proteins (A). qRT-PCR was conducted using primers targeting IBV gene 1 or the N gene to assess the levels of IBV genome and transcripts (B). The culture medium was collected and subjected to a plaque assay to measure the released virus titer (C). qRT-PCR was also used to measure the mRNA levels of IFNβ, IFITM3, and IL-8 (D). (E) DF-1 cells were transfected with RACK1 siRNA or NC for 48 h, followed by IBV infection for 2 h and subsequent transfection with poly(I:C), or treatment with TNFα, or IFNβ, respectively. Cells were harvested and subjected to qRT-PCR analysis to measure the mRNA levels of IFNβ, IFITM3, and IL-8. For panel A, the intensities of RACK1, p-PKCα, p-PKCβ, p-NUP62, and IBV N bands were normalized to β-actin, PKCα/β, NUP62, and β-actin, respectively. The ratio of these protein signals in RACK1 siRNA-transfected cells compared to NC-transfected cells is denoted as RACK1 (+:-), p-PKCα (+:-), p-PKCβ (+:-), p-NUP62 (+:-), and IBV N (+:-). For panels D and E, the value for mock-infected cells is set to 1. For panels B, D, and E, error bars represent the SD of technical triplicates. Statistical significance levels are denoted as follows: ns, P > 0.05; **P < 0.01; ***P < 0.001; ****P < 0.0001.

## N proteins from diverse coronaviruses promote the anchoring of p-PKCα to RACK1, and both PKCα/β and RACK1 are required for N protein to suppress the host antiviral response

One of the major functions of RACK1 is anchoring and trafficking activated PKC to specific subcellular locations [70]. To further investigate the effect of the interaction between IBV N and RACK1 on PKCα/β activity, Flag-tagged IBV N was expressed in DF-1 cells and immuno-precipitated using an anti-Flag antibody. Western blot analysis was then performed to detect endogenous RACK1, p-PKCα, and p-PKCβ. As shown in Fig 11A, endogenous RACK1 and p-PKCα were successfully co-immunoprecipitated with Flag-tagged IBV N, whereas p-PKCβ was not detected in the precipitated complex. This result indicates that IBV N, RACK1, and p-PKCα interact to form a complex.

In cells overexpressing HA-RACK1, a higher amount of p-PKCα, but not p-PKCβ, was co-immunoprecipitated with HA-RACK1 in the presence of IBV N (Fig 11B), indicating that IBV

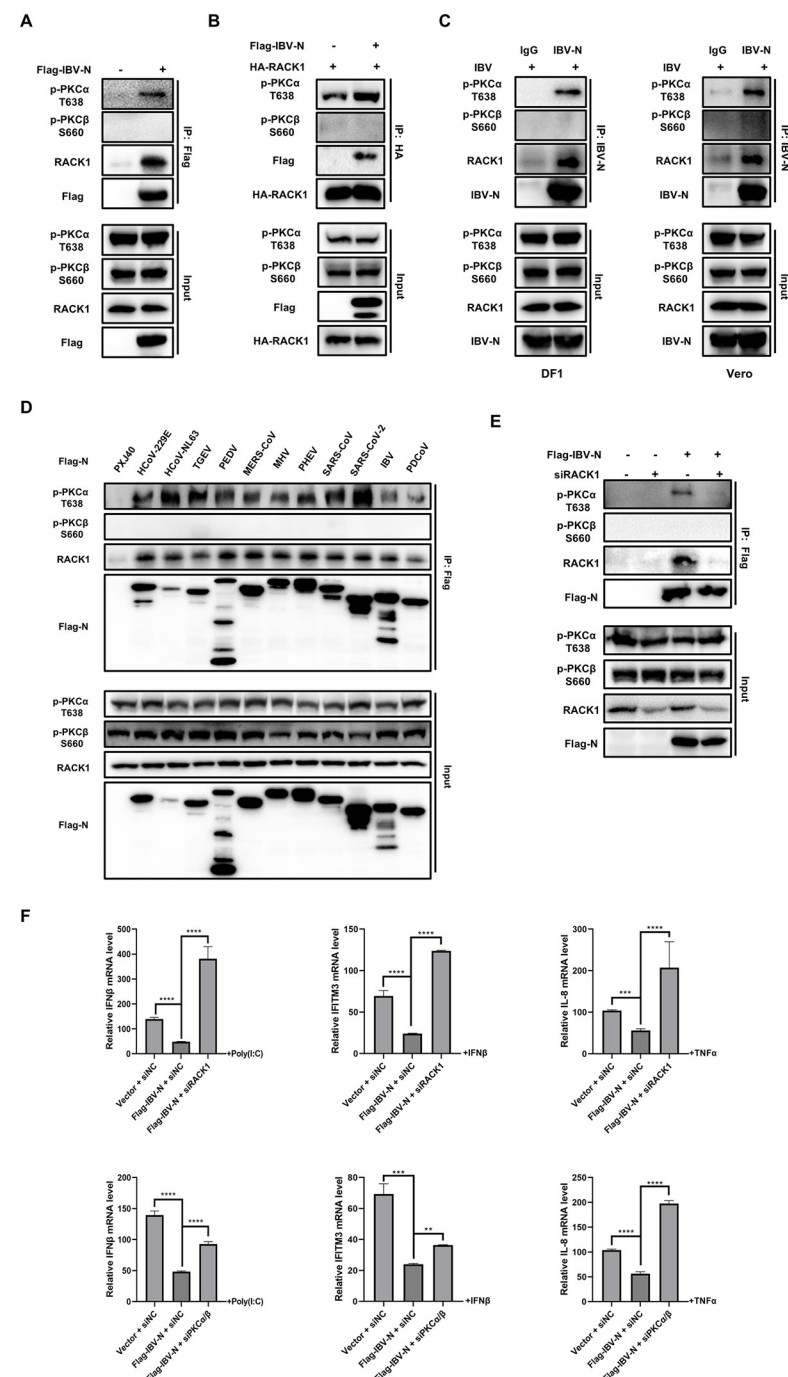

**Fig 11. Interaction between IBV N, p-PKCα, and RACK1, and their role in suppression of innate immune response.** (**A**) DF-1 cells were transfected with Flag-tagged IBV N or vector PXJ40 for 24 h. Cells were then subjected to Co-IP using anti-Flag antibody, followed by Western blot analysis. (**B**) Plasmids encoding HA-RACK1 and Flag-tagged IBV N, or HA-RACK1 and PXJ40, were co-transfected into DF-1 cells for 24 h. Cell lysates were subjected to Co-IP using anti-HA antibody, followed by Western blot analysis. (**C**) DF-1 cells or Vero cells were infected with IBV for 12 h and subjected to Co-IP using anti-IBV N antibody. The interactions of IBV N, p-PKCα, p-PKCβ, and RACK1 were detected by Western blot analysis. (**D**) HEK-293T cells were transfected with plasmids encoding Flag-tagged N proteins from eleven coronaviruses or PXJ40 for 24 h. Cell lysates were subjected to Co-IP with an anti-Flag antibody, followed by immunoblotting with the indicated antibodies. (**E**) siRACK1 or siRNA (NC) was transfected into DF-1 cells for 36 h, followed by transfection of plasmid encoding Flag-tagged IBV N or vector PXJ40 for 24 h. Cell lysates were subjected to Co-IP using anti-Flag antibody, followed by Western blot analysis. (**F**) siRNA targeting RACK1,

PKCα/β, or siRNA (NC) was transfected into DF-1 cells for 36 h, followed by transfection of plasmid encoding Flag-tagged IBV N or vector PXJ40. At 24 h post-transfection, cells were stimulated with poly(I:C), IFNβ, or TNFα. Cells were harvested 12 h post-stimulation, and the transcription levels of IFNβ, IFITM3, and IL-8 were measured by qRT-PCR analysis. The group transfected with vector PXJ40 without stimulation was set as 1. Error bars represent the SD of technical triplicates. Statistical significance levels are denoted as follows: **P < 0.01; ***P < 0.001; ****P < 0.0001.

N promotes the anchoring of p-PKCα to the scaffold protein RACK1. This interaction was further confirmed under IBV infection conditions, as shown by the successful pull-down of the N protein along with endogenous RACK1 and p-PKCα using an anti-IBV N polyclonal antibody, while p-PKCβ was not associated with RACK1 and the N protein (Fig 11C). These results validate the formation of a complex involving N, RACK1, and p-PKCα during IBV infection.

To determine whether the formation of the N-RACK1-p-PKCα complex is a conserved feature, Flag-tagged N proteins from eleven strains of coronaviruses were expressed in HEK-293T cells, followed by Co-IP using an anti-Flag antibody. HEK-293T cells were chosen for this study due to their high transfection efficiency, which facilitates the expression of N proteins. Western blot analysis revealed that all N proteins were able to bind and pull down endogenous RACK1 and p-PKCα; however, p-PKCβ was not detected in the precipitates (Fig 11D). These results confirm that the formation of the N-RACK1-p-PKCα complex is conserved across various coronavirus strains, and the N protein plays a critical role in promoting the anchoring of p-PKCα to RACK1.

Next, we investigated whether RACK1 acts as a scaffold to anchor N and PKCα together. RACK1 was knocked down in DF-1 cells, and then cells were transfected with IBV N. The Flag antibody successfully co-immunoprecipitated Flag-tagged IBV N and p-PKCα together with RACK1 in non-target control siRNA-transfected cells. However, this interaction was not observed in the absence of RACK1 (siRACK-transfected cells), where p-PKCα was not co-immunoprecipitated. As before, p-PKCβ was not co-immunoprecipitated with the IBV N protein (Fig 11E). These results confirm that RACK1 serves as the scaffold for the formation of the N-RACK1-p-PKCα complex.

To assess the impact of RACK1 and PKCα/β on the expression of antiviral genes, these proteins were individually knocked down in DF-1 cells, followed by overexpression of IBV N and stimulation with poly(I:C), IFNβ, or TNFα. The transcription levels of IFNβ, IFITM3, and IL-8 were examined by qRT-PCR. As shown in Fig 11F, knockdown of either RACK1 or PKCα/β significantly restored the expression of these antiviral genes, which had been suppressed by IBV N. These findings indicate that both RACK1 and PKCα/β are essential for the N protein to effectively suppress the innate immune response. The anchoring of p-PKCα to the RACK1-N complex, rather than p-PKCβ, along with the critical role of PKCα/β and RACK1 in NUP62 phosphorylation and N protein-mediated suppression of antiviral gene expression, suggests that the association with the RACK1-N complex enables p-PKCα to perform its kinase function in close proximity to its substrates, such as NUP62. In contrast, since p-PKCβ does not associate with the RACK1-N complex, it likely executes its kinase function through alternative mechanisms by translocating to specific subcellular locations.

## Nuclear export signal (NES) of IBV N is essential for promotion of cytoplasmic dispersion of FG-Nups and suppression of innate immune response

The coronavirus N protein compromises three highly conserved domains: the N-terminal viral RNA binding region (NTD), the Ser/Arg-rich region (SR-domain), and the C-terminal dimerization domain (CTD). To characterize which domain is involved inhibiting

nucleocytoplasmic trafficking, plasmids encoding four IBV N truncation fragments were constructed: ΔNTD with deletion of N-terminal 1 to 160 aa, the viral RNA binding region; ΔSR with deletion of the Ser/Arg-rich region 165 to 190 aa, the potential phosphorylation region; ΔCTD with deletion of the C-terminal 215 to 409 aa, the dimerization domain; ΔNES by removing the nuclear export signal (NES, [291]LQLDGLHL[298]) (Fig 12A). Vero cells were transfected with these plasmids and applied to immunofluorescence analysis. As shown in Fig 12B, both ΔNTD and ΔSR were distributed in the cytoplasm and exhibited the ability to inhibit the nuclear translocation of IRF3, p65, STAT1, and STAT2 in response to their respective stimuli, similar to wild-type IBV N. However, ΔCTD and ΔNES were primarily localized in the nucleus and showed a loss of capacity to inhibit the nuclear translocation of IRF3, p65, STAT1, and STAT2. The nuclear retention of ΔCTD might be attributed to the loss of NES ([291]LQLDGLHL[298]), which is located within the CTD (215 to 409 aa). Thus, the cytoplasmic distribution of N protein might be critical for perturbing the nucleocytoplasmic trafficking.

We further investigated the subcellular localization of RACK1 and PKCα/β in cells expressing various IBV N mutants. As illustrated in Fig 12C, cells transfected with PXJ40 exhibited RACK1 signal in the nucleus and perinuclear region, whereas PKCα/β signals were predominantly detected in the nucleus. In cells expressing wild-type IBV N, ΔNTD, or ΔSR, RACK1 and PKCα/β were redistributed to the cytoplasm, colocalizing with N protein or its truncated mutants. This indicates that N protein has the capacity to modulate the subcellular localization of RACK1 and PKCα/β. However, when ΔCTD and ΔNES mutants were confined to the nucleus, there was no significant dispersion of RACK1 and PKCα/β into the cytoplasm: RACK1 remained localized in the nucleus and perinuclear region, while PKCα/β predominantly remained in the nucleus (Fig 12C). These findings suggest that the localization of the N protein dictates the positioning of the RACK1-p-PKCα complex. Further examination of the impact of these N mutants on the intracellular distribution of Nups revealed that ΔNTD and ΔSR induced cytoplasmic dispersion of FG-Nups, NUP62, and NUP153, similar to cells expressing wild-type N protein. Conversely, in cells expressing ΔNES, FG-Nups, NUP62, and NUP153 signals remained concentrated at the nuclear envelope, with intense signals observed within the nucleus (Fig 12D). Notably, in cells expressing ΔCTD, a minor fraction of FG-Nups and NUP153 exhibited cytoplasmic dispersion, while the signal of NUP62 and NUP153 was intensified in the nucleus. Co-IP analysis revealed that similar to wild-type N protein, ΔNTD, ΔSR, ΔCTD, and ΔNES exhibited varying degrees of interaction capability with RACK1 and p-PKCα (Fig 12E).

Consistent with the observations in Fig 12B–12D, ΔNTD and ΔSR maintained the capability to inhibit the expression of IFNβ, IFITM3, and IL-8 in response to poly(I:C), IFNβ, and TNFα stimuli, whereas ΔCTD and ΔNES lost this capacity (Fig 12F). Overall, the presence of CTD and NES enables the N protein to localize in the cytosol alongside RACK1 and PKCα/β, thereby facilitating the function of the N-RACK1-p-PKCα complex in inducing cytoplasmic dispersion of FG-Nups. This process prevents the nuclear translocation of transcription factors and the subsequent antiviral innate immune response, as indicated by the results presented in Fig 12B–12F.

## Discussion

The coronavirus has evolved multiple strategies to inhibit the innate immune response for its own benefit. In this study, we demonstrate that IBV infection inhibits the expression of several antiviral genes by suppressing the nuclear translocation of their corresponding transcription factors: IRF3, STAT1/STAT2/IRF9, and p65. We identified the IBV N protein as the factor responsible for retaining these transcription factors in the cytoplasm and suppressing the

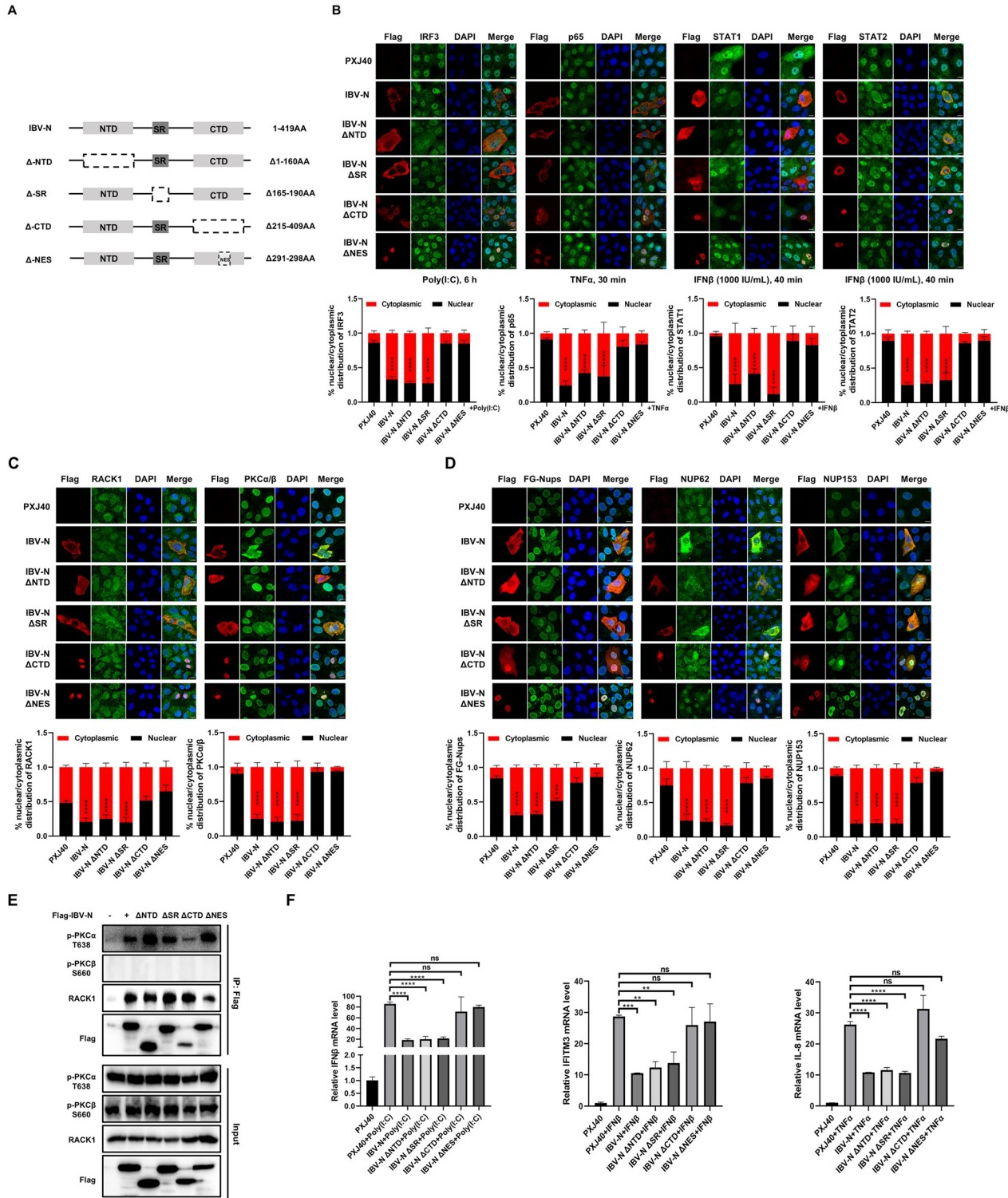

**Fig 12. Significance of cytoplasmic localization of IBV N protein in disrupting the nuclear envelope localization of Nups and suppressing antiviral gene expression.** (A) Schematic representation of truncated mutants of the IBV N protein. (B) Vero cells were transfected with plasmids encoding Flag-tagged IBV N, ΔNTD, ΔSR, ΔCTD, ΔNES, or PXJ40. At 18 h post-transfection, cells were treated with poly (I:C), while treatment with IFNβ or TNFα was initiated at 24 h post-transfection. Subsequently, cells were subjected to immunofluorescence analysis. (C-D) Vero cells were transfected with plasmids encoding Flag-tagged IBV N, ΔNTD, ΔSR, ΔCTD, ΔNES, or PXJ40. At 24 h post-transfection, cells were subjected to immunofluorescence analysis. (B-D) Representative images from three independent experiments are shown, with scale bars indicating 10 μm. The

fluorescence signals of corresponding proteins in in vector-transfected and N expressing cells from three fields of view were quantified using ImageJ. The intensities of the fluorescence signals in the nucleus (black bars) and the cytoplasm (red bars) are presented as bar graphs, with error bars representing the SD. (E) DF-1 cells were transfected with plasmids encoding Flag-tagged IBV N or the truncated mutants, or PXJ40. At 24 h post-transfection, cell lysates were subjected to Co-IP with anti-Flag antibody and subsequently immunoblotted with corresponding antibodies. (F) DF-1 cells were transfected with plasmids encoding Flag-tagged IBV N, its truncated mutants, or vector PXJ40. At 24 h post-transfection, cells were treated with poly(I:C), IFNβ, or TNFα for 12 h. Cells were harvested, and the levels of IFNβ, IFITM3, and IL-8 were determined using qRT-PCR analysis. Error bars represent the SD of technical triplicates. Statistical significance is indicated as follows: ns, P > 0.05; * P < 0.05; ** P < 0.01; *** P < 0.001; **** P < 0.0001.

expression of antiviral genes. Both IBV infection and N protein expression promote the cytoplasmic dispersion of multiple Nups, indicating perturbation of the NPC function, which governs the nucleocytoplasmic trafficking of RNA and proteins. Although immunofluorescence analysis shows the colocalization of the N protein and Nups in the cytoplasm, Co-IP analysis indicates no direct interaction between the N protein and several Nups, including NUP62 and NUP42 (S7 Fig). Previous studies have reported that the disassembly of the NPC is regulated by the phosphorylation of Nups during cell mitosis [56, 71]. In this study, Western blot analysis demonstrates that NUP62 is phosphorylated at residues T269 and S272 during IBV infection. Chemical inhibition of intracellular PP1 and PP2A activities using okadaic acid results in the accumulation of phosphorylated NUP62 and promotes its cytoplasmic dispersion. These findings strongly support the idea that phosphorylation events of Nups lead to the disassembly of the NPC. Mechanistic studies reveal that the N protein interacts with RACK1 and promotes the anchoring of p-PKCα, but not p-PKCβ, to RACK1. The presence of both RACK1 and PKCα/β is required for the phosphorylation of NUP62 and suppression of antiviral gene expression, thereby benefiting virus infection. Inhibition of PKCα/β activity by chemical inhibitor prevents the cytoplasmic distribution of multiple Nups in both N protein-expressing or IBV-infected cells, further indicating the role of PKCα/β in Nups phosphorylation and NPC disassembly. Although phosphorylation of other Nups was not detected due to the lack of phospho-specific antibodies, the cytoplasmic dispersion of FG-Nups, NUP42, NUP153, and TPR suggests that additional Nups might undergo phosphorylation during IBV infection. These observations were also apparent in cells expressing the N proteins across various coronaviruses. Hence, coronavirus infection potentially triggers the phosphorylation of Nups and the disassembly of the NPC through the N-RACK1-p-PKCα-Nup signaling axis. Disruption of the NPC hinders transcription factors from accessing the nucleus, consequently inhibiting the transcription of antiviral genes and ultimately leading to immune suppression, thereby favoring viral replication (Fig 13).

FG-Nups primarily anchor to the central channel of the NPC, forming a dense barrier that prevents passive diffusion while facilitating the passage of cargos via nuclear transport receptors [72]. During IBV infection, no degradation of FG-Nups is observed; however, NUP42 exhibits a shift in size, and phosphorylation of NUP62 at T269 and S272 is confirmed by anti-p-NUP62 antibody (Fig 2C). Extensive phosphorylation of Nups has been reported to disrupt protein-protein interactions at key contact nodes within the NPC, leading to NPC disintegration and dispersion of Nups into the cytosol [56]. Although we were unable to detect phosphorylation of NUP98 by Western blot analysis due to the lack of an anti-p-NUP98 antibody, immunofluorescence analysis revealed the breakdown of the NUP98 nuclear ring signal in cells expressing N proteins, indicating disassembly of this Nup (Fig 5B). The central FG-Nup subunit NUP62 is demonstrated to be phosphorylated and disperses into the cytosol, along with the cytoplasmic dispersion of Nup153, Nup42, and TPR (Fig 2A and 2C). When the PP1 and PP2A inhibitor okadaic acid was applied to cells, the phosphorylation level of NUP62 was dramatically increased (S6A Fig), which coincided with cytoplasmic dispersion (S6B Fig). This

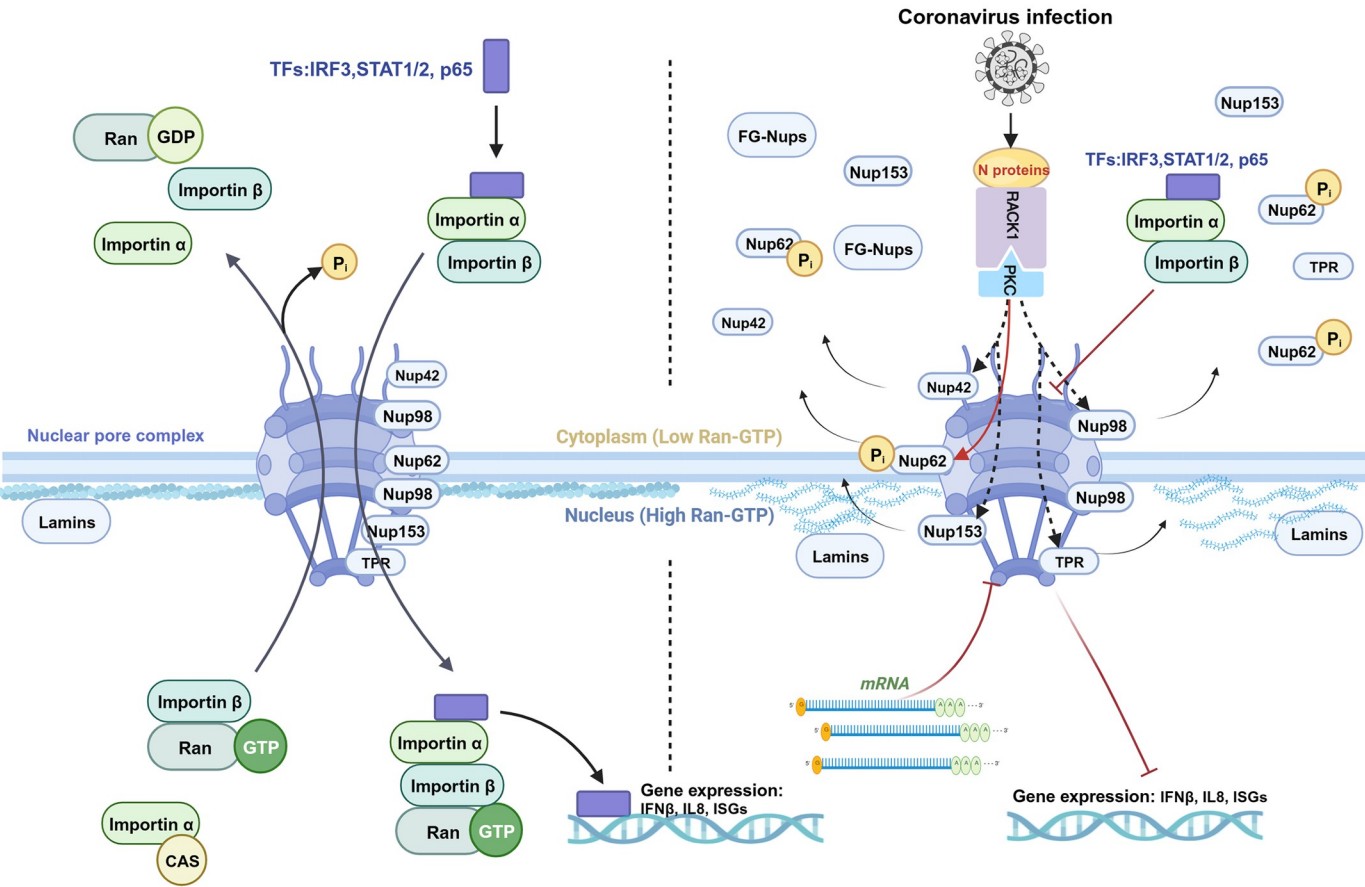

**Fig 13. A working model illustrating how coronavirus N protein disrupts the nuclear transport system to inhibit the nuclear translocation of transcription factors and subsequent gene expression.** In the left panel, the canonical nuclear import pathway is described, where the NLS of cargo proteins (e.g., transcription factors) is recognized by nuclear import receptors such as importin α, which forms a complex with importin β. This import complex translocates through the NPC by interacting with FG-Nups. Upon entering the nucleoplasm, importin β binds to Ran-GTP, leading to the disassembly of the import complex and the release of the cargo. Importin β, bound to Ran-GTP, is then transported back to the cytoplasm, while importin α is recycled by the cellular apoptosis susceptibility (CAS) protein (also known as exportin 2). Hydrolysis of GTP by Ran releases importin β for the next round of import. In the right panel, during coronavirus infection, the N protein interacts with RACK1 and recruits p-PKCα to form a ternary N-RACK1-p-PKCα complex. This complex phosphorylates NUP62, causing its cytoplasmic dispersion along with other Nups. As a result, the NPC becomes unable to transport cargo into the nucleus, thereby inhibiting the nuclear translocation of transcription factors such as IRF3, STAT1/STAT2, and p65. This inhibition subsequently blocks the expression of downstream antiviral genes. This figure was created using Biorender (BioRender, Toronto, ON, Canada).

observation demonstrates a direct correlation between the phosphorylation of NUP62 and its diffusion into the cytosol. It is noteworthy that in cells treated with okadaic acid, the cytoplasmic distribution pattern of FG-Nups (S6B Fig) is not as pronounced as in IBV-infected or N protein-expressing cells; most FG-Nups remain associated with the nuclear envelope. Thus, in virus-infected cells or cells expressing the N protein, the NPC is dismantled more extensively than in cells treated with a PP1 and PP2A inhibitor. Moreover, inhibition of phosphatase activity by inhibitor results in the appearance of two p-NUP62 bands around 120 kDa and 62 kDa (S6A Fig), indicating the formation of dimers after hyperphosphorylation. The antibody against total NUP62 detects three bands adjacent to 62 kDa in inhibitor-treated cells (S6A Fig), suggesting that NUP62 harbors multiple phosphorylation sites in addition to T269 and S272. It has been reported that extracellular signal-regulated kinase (ERK) and p38 MAPK, activated by encephalomyocarditis virus leader protein, are involved in hyperphosphorylation of several FG-Nups including NUP62, NUP153, and NUP214, leading to diffusion of their

nuclear ring signals and inhibition of nuclear import [73,74]. The ERK-targeted phosphorylation site of NUP62 has previously been mapped to a single PxTP motif within the FG repeat region of NUP62, resulting in an alteration in NPC sensitivity to STAT3 passage [75]. In this study, we identified the phosphorylation sites of NUP62 at residue T269 and S272 within its flexible region. Hyperphosphorylation of NUP62 was observed to coincide with the cytoplasmic dispersion of NUP62 itself and other Nups, as well as the inhibition of transcription factors import. Based on these findings, we conclude that the disruption of NPC integrity in IBV-infected cells is attributed to phosphorylation events involving NUP62 and other FG-Nups. Additionally, it is worth noting that Nup42 exhibited a size shift during IBV infection (Fig 2C). Investigating whether this phenomenon impacts host nucleocytoplasmic transport is one of our next research directions.

It has been elucidated that the disintegration of the nuclear envelope induced by parvovirus infection involves a sequential enzymatic cascade mediated by PKC, CDK2, and caspase-3 [76]. In this study, we focused on screening the kinases responsible for NUP62 phosphorylation using chemical inhibitors, which prompted us to investigate PKCα/β. PKCα/β is known for its ability to phosphorylate a diverse set of protein substrates and its involvement in various cellular processes, including cell adhesion, cell transformation, cell cycle regulation, apoptosis, and macrophage development [77]. Several direct substrates of PKCα/β have been identified, such as RAF1 [78], BCL2 [79], DOCK8 [80], Lamin B1 [63] and components of the signaling cascade involving ERK1/2 [81], as well as RAP1GAP [82]. In our study, we observed that the PKCα/β specific inhibitor, Enzastaurin, effectively suppresses the cytoplasmic dispersion of several Nups in both IBV-infected cells and cells expressing the N proteins from diverse coronaviruses (Figs 8A–8C and 9C), indicating a clear correlation between PKCα/β activity and NPC disassembly. Inhibition of PKCα/β activity or knockdown of PKCα/β specifically reduces the level of phosphorylated NUP62 (Fig 9D and 9G). This study provides the first evidence that PKCα/β is involved in the phosphorylation of NUP62 and its subsequent cytoplasmic dispersion. Moreover, the presence of PKCα/β is crucial for the suppression of antiviral gene expression mediated by IBV or its N protein (Figs 9J and 11F) and also facilitates IBV infection (Fig 9D–9I). Supporting our findings, a recent study demonstrated that the replication of SARS-CoV-2 is inhibited by pan-PKC inhibitors such as Go 6983, Bisindolmaleimide I, Enzastaurin, and Sotrastaurin [83], suggesting that PKC plays a critical role in facilitating coronavirus infection. Therefore, targeting PKC presents a promising strategy for developing broad-spectrum anti-coronaviral drugs.

The regulation of PKC signaling by coronavirus involves the interaction of the viral N protein with the PKC scaffold protein RACK1, as revealed by N protein interactome analysis and Co-IP. As a highly conserved multifunctional protein, RACK1 interacts directly or in complex with various cellular proteins, including PKCα/βII, contributing to protein shuttling, subcellular localization, and activity modulation [62, 84]. In IBV-infected or diverse coronaviruses N proteins-expressing cells, we observed an augmentation in the interaction between p-PKCα and RACK1 (Fig 11A–11D), with PKCα/β and RACK1 exhibiting colocalization with the N protein in the cytoplasm (Figs 7C and 9B). These findings suggest that the N protein facilitates the translocation of p-PKCα from the nucleus to the cytoplasm, where it associates with RACK1. The depletion of RACK1 demonstrates its indispensability in the formation of the N-RACK1-p-PKCα complex (Fig 11E). Furthermore, the presence of RACK1 is essential for the phosphorylation of PKCα, PKCβ, and NUP62, as well as for efficient IBV replication (Fig 10A-C). The conserved enhancement of p-PKCα anchoring to RACK1 by the N protein across diverse coronaviruses suggests a common mechanism employed by coronaviruses to recruit p-PKCα to RACK1, promoting the phosphorylation and cytoplasmic dispersion of NUP62 (Fig 11D), as well as potentially affecting other Nups. The necessity of RACK1 and PKCα/β for the

N proteins or IBV to suppress the expression of antiviral factors such as IFNβ, IFITM3, and IL-8 (Figs 9J, 10E, and 11F) further underscores the critical role of the N-RACK1-p-PKCα complex in antagonizing the host innate immune response, potentially through phosphorylation events on NUP62 or other substrates. In line with our findings, RACK1 has been implicated in facilitating SARS-CoV-2 replication, with studies showing that its depletion reduces infectious virus release and intracellular spike protein expression [85]. Additionally, it has been demonstrated that the N protein of porcine reproductive and respiratory syndrome virus (PRRSV) interacts with RACK1, thereby promoting PRRSV replication [86]. This suggests that RACK1 may play a pivotal role not only in coronavirus infections but also in arterivirus infections. Previous research indicates that part of the RACK1-binding site for PKCβII resides within the PKCβII V5 domain, and a peptide corresponding to 645–650 aa in PKCβII selectively inhibits phorbol 12-myristate 13-acetate (PMA)-induced translocation of PKCβII, thereby blocking PKC activity [87]. Developing RACK1-competitive PKC inhibitors could therefore represent a novel strategy for creating anti-coronaviral therapeutics.

As a multifunctional protein, the coronavirus N protein plays pivotal roles in packaging viral RNA into ribonucleoprotein, participating in virion assembly, modulating viral replication and transcription, arresting cell cycle and regulating host innate immunity [88, 89]. Our study reveals that coronaviruses N proteins interact with RACK1 and p-PKCα (Fig 11D), as well as induce the cytoplasmic dispersion of multiple Nups (Fig 5A–5B), leading to the blockade of nuclear translocation of transcription factors such as IRF3, STAT1/STAT2/IRF9, and p65 (Fig 5D–5E), ultimately inhibiting the expression of antiviral genes (Fig 5G). The N protein consist of three conserved domains: NTD, SR domain, and CTD [90–92]. Using IBV N protein as a model, we identified the NES residing in NTD as the primary sequence responsible for the cytoplasmic distribution of the N-RACK1-p-PKCα complex and Nups, prevention of the nuclear import of transcription factors, and repression of antiviral gene expression (Fig 12). Our findings suggest that within the ternary complex of IBV N, RACK1, and p-PKCα, N plays a pivotal role in determining the subcellular localization of the complex. The full-length N protein, ΔNTD, or ΔSR, interact with RACK1 and p-PKCα, causing their distribution in the cytoplasm. However, N protein lacking CTD or NES is retained in the nucleus together with RACK1 and PKCα/β, thereby losing the ability to promote the cytoplasmic dispersion of Nups and subsequent nuclear importing and transcription events. Thus, the cytoplasmic localization of IBV N determines the positioning and regulation role of the N-RACK1-p-PKCα complex in phosphorylating NUP62 or other substrates. We attempted to generate a recombinant virus by deleting the NES of the N protein in the IBV Beaudette strain using reverse genetics. However, we were unable to rescue the NES-deficient rIBV strain. This underscores the critical importance of nuclear export and cytoplasmic localization for the N protein to fulfill its function, which is indispensable for IBV replication. Although limited research has been conducted on the NES of N proteins from other coronaviruses, nucleolar localization appears to be a common feature across the N proteins of all four genera of coronaviruses [93, 94]. This suggests that the N protein possesses the ability to shuttle between the cytoplasm and nucleus; for instance, phosphorylated SARS-CoV N protein is translocated from the nucleus to the cytoplasm with the assistance of 14-3-3 proteins [95]. Interestingly, the N protein is a key factor in coronavirus-mediated cell cycle regulation, which creates a more favorable environment for viral replication, thereby enhancing replication efficiency [96]. Previous studies have shown that the N proteins of SARS-CoV, PEDV, and TGEV induce host cell cycle arrest in the S phase by suppressing the activity of cyclin-CDK complexes [97–99]. Additionally, the IBV N protein may regulate the cell cycle by interacting with nucleolin, a nucleolus-localized protein implicated in cell growth and cell cycle regulation [100]. Given that hyperphosphorylation and cytoplasmic dispersion of key Nups are crucial steps in the cell

cycle, and that the interaction between the N protein, p-PKCα, and RACK1 promotes these processes, the potential role of the N-RACK1-p-PKCα interaction in cell cycle regulation warrants further investigation.

The disassembly of the NPC not only inhibits the nuclear transport of transcription factors involved in the immune response but also disrupts host mRNA export and sequesters nuclear proteins essential for viral replication in the cytoplasm, thereby facilitating viral replication. In addition to inhibiting the nuclear translocation of STAT1, a recent study demonstrated that SARS-CoV and SARS-CoV-2 ORF6 interacts with NUP98/Rae1 to impede cellular mRNA export, reducing the translation of antiviral genes and diverting the limited cellular translational machinery toward viral translation [101]. In our study, N protein-expressing cells exhibited nuclear retention of mRNA signals and reduced protein synthesis (S5 Fig). Future investigations are warranted to elucidate the mechanisms by which the N protein prevents host mRNA export, reducing host translation events, or retains nuclear proteins in the cytoplasm to facilitate virus replication.

In addition to the N protein, the screening study presented in Fig 3 identified that IBV nsp3, nsp6, nsp7, nsp9, E, M, and 5a altered the localization of FG-Nups. Nsp3 and nsp6 are primarily localized to the ER, where they may induce membrane curvature, contributing to the formation of double-membrane vesicles [102–106]. Similarly, the M protein and E protein induce the ER-Golgi intermediate compartment (ERIGC) membrane curvature and assist the assembly and budding of infectious virions [88, 107]. Given that the endoplasmic reticulum is contiguous with the nuclear envelope, which contains the NPC [10], the membrane manipulation abilities of these viral proteins may lead to the mislocalization of FG-Nups. The underlying mechanisms by which these viral proteins alter the nuclear membrane localization of FG-Nups warrant further investigation. Fig 3 demonstrated that nsp2, nsp5, nsp6, nsp12, nsp13, nsp15, E, M, 5a and N blocked IFNβ-induced STAT1 nuclear translocation or reduced STAT1 signaling, consistent with the suppression of IFITM3 expression by these viral proteins, as shown in S4 Fig. This inhibition may result from interference with the nuclear transport system or direct targeting of the JAK-STAT signaling pathway. Additionally, in S4 Fig, we observed that nsp8 and nsp9 significantly enhanced the expression of IFNβ or IFITM3 in response to relevant stimuli, indicating that these proteins stimulate innate immune signaling pathways. The primer extension activity of nsp8 may produce short RNA fragments that enhance the cellular IFN response [108, 109]. Consistent with our findings, reports indicate that SARS-CoV-2 nsp9 promotes cytokine production by interacting with and activating TBK1 [110]. Furthermore, S4 Fig revealed that IBV nsp2, nsp3, nsp6, nsp7, nsp8, nsp9, nsp12, nsp13, and nsp14 enhanced the activation of the NF-κB signaling pathway, demonstrating that these viral proteins can stimulate inflammatory responses. It is well established that coronavirus infections can trigger inflammatory responses and cause severe tissue damage, as seen in IBV [111], PEDV [112], SARS-CoV-2 [113,114], MERS-CoV [115], and SARS-CoV [116–118]. Reports indicate that both SARS-CoV-2 and TGEV nsp2 induce NF-κB-driven inflammatory responses [119,120]. The evolutionarily conserved C-terminal domain (CoV-Y) of SARS-CoV-2 nsp3 has been shown to interact with BRAP, enhancing the phosphorylation of IκBα and IκBβ, thereby stimulating NF-κB signaling and inducing host inflammatory responses [121]. Another study demonstrated that SARS-CoV-2 nsp6 interacts with TAK1, where TRIM13-mediated K63 polyubiquitination of nsp6 at K61 is crucial for recruiting NEMO to the nsp6-TAK1 complex and its subsequent activation [122]. Additionally, the nsp7 +nsp8 complex acts as a unique multimeric RNA polymerase capable of both *de novo* initiation and primer extension [108,109]. The synthesis of short RNA by this complex may further contribute to the activation of NF-κB signaling. Several reports indicate that SARS-CoV-2 nsp14 activates NF-κB signaling by associating with host inosine-5'-monophosphate dehydrogenase

2 (IMPDH2) [123], promoting IKK phosphorylation [124], and that its MTase activity is critical for inducing canonical NF-κB activation [125]. Overall, the ability of multiple viral proteins to regulate innate immune response and inflammatory response pathways may provide the virus with a survival advantage, allowing it to adapt to variations in host responses.

In summary, our study reveals a novel mechanism by which the IBV N protein promotes the anchoring of p-PKCα to RACK1 and subsequently relocates the N-RACK1-p-PKCα complex to the cytoplasm. In this context, p-PKCα phosphorylates NUP62 and potentially other Nups, leading to their disassembly and cytoplasmic dispersion. This process effectively inhibits the nuclear import of transcription factors such as IRF3, STAT1/STAT2/IRF9, and p65, which blocks the expression of antiviral genes and ultimately facilitates IBV replication. Previous studies have shown that other coronaviruses, such as SARS-CoV, SARS-CoV-2, and MERS-CoV, utilize different strategies, like the accessory protein ORF6 and ORF4b, to inhibit nuclear transport [50–52]. Our findings complement this body of work by demonstrating that coronaviruses share common strategies for evading immune responses. The disruption of nuclear trafficking and inhibition of critical transcription factors' nuclear entry by the N protein represent a novel and evolutionarily conserved function across diverse coronaviruses. By inhibiting the nuclear import of transcription factors crucial for antiviral gene expression, the N protein effectively dampens the host's immune response, potentially leading to more severe or prolonged infections. This disruption of nucleocytoplasmic trafficking by the N protein opens up potential therapeutic avenues. The conserved nature of the N protein's function across coronaviruses underscores the importance of this mechanism in viral pathogenesis and offers valuable insights for developing new strategies to combat coronavirus infections.

## Materials and methods

### Cells and viruses

Chicken embryo fibroblast DF-1 cells (CRL-3586), African green monkey kidney epithelial Vero cells (CRL-1586), and human embryonic kidney HEK-293T cells (CRL-3216) were obtained from ATCC. These cells were cultured in Dulbecco's Modified Eagle Medium (DMEM) supplemented with 10% (v/v) fetal bovine serum (FBS) (Gibco-Thermo Fisher, Waltham, MA, USA). The IBV Beaudette strain was kindly provided by Prof. Dingxiang Liu's laboratory at South China Agricultural University.

### Antibodies and chemicals

Rabbit anti-IBV-N and mouse anti-IBV-N polyclonal antibodies were generated in our laboratory. Additionally, the following antibodies were purchased: rabbit anti-IRF3 (ab68481), rabbit anti-p65 (ab32536), rabbit anti-p38 (ab170099), rabbit anti-IRF-9 (ab271043), rabbit anti-TPR (ab170940), rabbit anti-Ran (ab157213), rabbit anti-NUP42 (Nucleoporin hCG1, ab192609), rabbit anti-phospho-NUP62 (T269 + S272) (ab183480), rat anti-NUP62 (ab188413), mouse anti-NUP153 (ab24700), mouse anti-FG-Nups [Mab414] (ab24609), mouse anti-importin β1 (ab2811), rabbit anti-PKCα/βII (ab184746), rabbit anti-phospho-PKCα (T638) (ab32502), rabbit anti-PABP (ab21060), rabbit anti-NUP98 (#2598), rabbit anti-phospho-PKCβII (Ser660) (#9371), rabbit anti-STAT1 (#14994), rabbit anti-phospho-STAT1 (#9167), rabbit anti-phospho-STAT2 (#88410), rabbit anti-phospho-p65 (#3033), rabbit anti-phospho-IRF3 (#4947), rabbit anti-phospho-p38 (#9211), rabbit anti-HA Tag (#3724), rabbit anti-Flag Tag (#14793), rabbit anti-PP1α (#2582), rabbit anti-PP2A C Subunit (#2038), rabbit anti-STAT2 (16674-1-AP), rabbit anti-NUP62 (13916-1-AP), rabbit anti-importin α1 (16674-1-AP), mouse anti-puromycin (MABE343) and rabbit anti-RACK1 (R1905) were purchased from Abcam, Cell Signaling Technology, Proteintech Group, and Merck, respectively. Chicken anti-Flag Tag

(AFLAG) was purchased from Exalpha. Mouse anti-Flag Tag (M185-3L) was purchased from MBL, and rabbit anti-β-actin (AC026) was also used. The dilution of antibodies and their cross-reactivity with corresponding chicken proteins are summarized in S1 Table. Alexa Fluor goat anti-rabbit-488 (A-11034), Alexa Fluor goat anti-rabbit-594 (A-11037), Alexa Fluor goat anti-mouse-488 (A-11029), and Alexa Fluor goat anti-mouse-594 (A-11005) were obtained from Invitrogen, USA. Poly(I:C) (31852-29-6) and Puromycin (ant-pr-1) were purchased from InvivoGen, France. Recombinant human IFN-β protein (#8499-IF) was purchased from Bio-Techne R&D Systems, USA. Recombinant Human TNF-α (P00029) was purchased from Solarbio, China. Okadaic acid (#5934) was purchased from Cell Signaling Technology, USA. Enzastaurin (HY-10342) was purchased from MCE, China. The ClonExpress Ultra One Step Cloning Kit (C115) and Mut Express II Fast Mutagenesis Kit V2 (C214) were purchased from Vazyme, China. The biotinylated Oligo(dT) Probe was purchased from Promega, USA.

## Plasmids construction

Plasmids encoding HCoV-229E-N, HCoV-NL63-N, TGEV-N, PEDV-N, MERS-CoV-N, MHV-N, PHEV-N, SARS-CoV-N, and SARS-CoV-2-N were provided by Prof. Tongling Shan (Shanghai Veterinary Research Institute, CAAS) [126]. Plasmids encoding Flag-tagged IBV nsp2, nsp3, nsp5, nsp6, nsp7, nsp8, nsp9, nsp12, nsp13, nsp14, nsp15, nsp16, E, M, 5a, and N, constructed by Dr. Gao, are maintained in our laboratory [127]. The plasmid encoding IBV N was generated by amplifying cDNA from IBV Beaudette-infected DF-1 cells using corresponding primers and cloning into PXJ40. IBV N ΔNTD, ΔSR, ΔCTD, and ΔNES mutants were generated by mutagenesis of the Flag-tagged IBV N plasmid using the Mut Express II Fast Mutagenesis Kit V2. PKCα and PKCβ genes were synthesized and ligated into the pCMV-HA expression vector by Sangon Bioengineering (Shanghai) Co., Ltd., Shanghai, China. NUP62, NUP42, and RACK1 genes were cloned by RT-PCR from HEK-293T cells and ligated into the pCMV-HA expression vector using the ClonExpress Ultra One Step Cloning Kit. The SV40 large T antigen NLS sequence [128] was incorporated into the forward primer used for amplifying the EGFP gene. The EGFP gene was then cloned by PCR from the pEGFP-N1 expression vector and subsequently ligated back into the pEGFP-N1 expression vector using the ClonExpress Ultra One Step Cloning Kit, resulting in the creation of the 2 × EGFP-NLS expression vector. The corresponding primers used to generate the above plasmids are listed in S2 Table.

## Cell transfection and RNA interference

Vero cells or DF-1 cells were seeded in 6-well plates, 12-well plates, or chamber slides (Nunc Lab-Tek II Chamber Slide System, Thermo Fisher Scientific, USA) with 70–80% confluency. The indicated plasmids were transfected into cells using Lipofectamine 2000 (Invitrogen, Carlsbad, CA) according to the manufacturer's instructions. Briefly, 1 μg of plasmid and 3 μL of Lipofectamine 2000 (m/v = 1:3) were diluted in 0.1 mL of Opti-MEM (Gibco, 31985070, Gaithersburg, MD). After 5 min of incubation, the plasmid and Lipofectamine 2000 were mixed and incubated at room temperature for 20 min to allow the formation of lipid-plasmid complexes. Finally, the complexes were added to the cultured cells and incubated for 24 h.

To knock down chicken RACK1 and PKCα/β genes in DF-1 cells, siGenome Gallus gallus RACK1 and PKCα/β siRNA were purchased from GenePharma Co, China. The sequences targeting RACK1 and PKCα/β were as follows: RACK1-siRNA: 5'-CGGGAUAUCUGAACACA-GUTT-3'; PKCα/β-siRNA: 5'-GGAGCUCUAUGCA AUCAAATT-3'. A non-targeting control siRNA was also provided by GenePharma Co and used as a control with no specific gene targeting. For each siRNA transfection, 100 pmol of siRNA and 5 μL of Lipofectamine 2000 were diluted in 0.1 mL of Opti-MEM, respectively. After 5 min of incubation, the siRNA

and Lipofectamine 2000 were mixed and incubated at room temperature for 20 min, allowing the formation of lipid-siRNA complexes. The complexes were then added to the cultured cells (30–40% confluency) and incubated for 48 h, followed by plasmid transfection or IBV infection. Cells were subjected to Western blot analysis, real-time qRT-PCR analysis, or plaque assay at the indicated time points.

## Plaque assay

To measure infectious virus titer, a standard plaque assay was performed as follows: Vero cells were seeded in 6-well plates and inoculated the next day with 10-fold serial dilutions ($10^{-1}$ to $10^{-6}$) of the culture medium containing the virus particles in a total volume of 2 mL of DMEM. After 90 min of adsorption at 37˚C, the cells were overlaid with 1% methylcellulose (Sigma-Aldrich, M0430, Germany) diluted in 2 × DMEM medium. Three days later, the plates were fixed with 4% formaldehyde for 1 h at room temperature. After the removal of methylcellulose, the fixed cells were stained with 0.1% crystal violet (Beyotime, C0121, China) for 30 min at room temperature. Excess crystal violet was rinsed off with PBS, and the stained cell plates were air-dried overnight at room temperature. Plaques were counted the next day. To minimize error, only wells containing between 10 and 100 plaques were counted. The virus titer was calculated using the following equation: plaque forming unit (PFU)/mL = Number of plaques /inoculated volume of the virus (mL) × virus dilution.

## Western blotting analysis

DF-1 cells or Vero cells were seeded in 6-well plates and transfected with various siRNAs or infected with IBV at an MOI of 1 according to experimental requirements. Cells were harvested at the indicated time points or treated with Enzastaurin (1μM, 16 h) or okadaic Acid (1μM, 1 h), with DMSO included in the parallel experiment as a negative control. Cell samples were lysed in 2 x protein loading buffer (20 mM Tris-HCl, 2% SDS, 100 mM DTT, 20% glycerol, 0.016% bromophenol blue) and incubated in a 100˚C metal bath for 10 min to fully denature the proteins. The denatured cell samples were then subjected to centrifugation at 12,000 rpm for 5 min. The supernatant proteins were resolved on a 10% SDS-PAGE and transferred to a nitrocellulose membrane (0.45 μm, Millipore, USA). Membranes were blocked in blocking buffer (5% nonfat milk, TBS, 0.1% Tween 20) for 1 h, followed by overnight incubation at 4˚C with primary antibodies diluted in dilution buffer (Beyotime, P0023, China) as indicated in S1 Table. The membranes were then incubated with secondary antibodies conjugated with HRP (Invitrogen, USA) diluted 1:10,000 in blocking buffer for 1 h at room temperature. After each incubation, membranes were washed three times with washing buffer (0.1% Tween in TBS). Proteins were visualized using the ECL detection system (Thermo, Rockford, IL). ImageJ program (NIH, USA) was used to quantify the intensities of corresponding bands on the Western blot according to the manufacturer's instructions.

## Indirect immunofluorescence analysis

Cells were seeded onto chamber slides and transfected with various plasmids or infected with IBV at an MOI of 1, according to experimental requirements. At the indicated time points, cells were transfected with poly (I:C) (20 μg/mL) or treated with IFNβ (1000 IU/mL, 40 min), TNFα (20 ng/mL, 30 min), UV irradiation (1.92 J/cm2, 20 min), Enzastaurin (1 μM), okadaic acid (1 μM, 1 h), or puromycin (5 μg/mL, 1 h). DMSO was used as the negative control in the case of drug treatment. Following treatment, cells were fixed with 4% paraformaldehyde for 15 min at room temperature. After three washes with PBS, cells were permeabilized with 0.5% Triton X-100 for 15 min and incubated in blocking buffer (3% BSA in PBS) for 1 h. Cells were

then incubated with the primary antibody diluted in blocking buffer for 2 h at 37˚C (the dilution was indicated in S1 Table), followed by incubation with Alexa Fluor-conjugated secondary antibody diluted 1:500 in blocking buffer for 1 h at 37˚C. In the case of double staining, cells were further incubated with the other primary antibody, followed by incubation with the corresponding fluorescent-conjugated secondary antibody. After each incubation step, cells were washed three times with PBST. DAPI (Beyotime, C1002, China) was then applied to stain the nuclei for 10 min. Finally, cells were washed three times with PBST, and the subcellular localization of corresponding proteins was examined using a Zeiss LSM880 confocal microscope.

Three confirmatory experiments were conducted, and the representative images were shown. The ImageJ program (NIH, USA) was used to quantify the intensities of the protein signals. To assess the nuclear/cytoplasmic distribution of the protein of interest, DAPI-stained nuclei were used as masks for quantifying nuclear intensity. Given that nucleoporins and activated transcription factors are primarily localized in the nucleus, while the coronavirus N protein is found in the cytoplasm, we defined the entire cellular area based on the distribution of the N protein. This allowed us to calculate the total fluorescence intensity of the target proteins within the whole cell. We collected the total cellular fluorescence and subtracted the nuclear fluorescence of the target protein to obtain the cytoplasmic fluorescence. For each analyzed cell, the nuclear/cytoplasmic fluorescence distribution of different proteins was calculated by dividing the nuclear or cytoplasmic fluorescence by the total cell fluorescence. Statistical analysis was conducted using GraphPad Prism 8 software. The quantification is based on three fields of view within a single experiment, with error bars representing standard deviation (SD) in the histogram of nuclear/cytoplasmic fluorescence distribution for the protein of interest. The significance of differences between two groups was evaluated using a two-tailed independent Student's t-test. A p-value of less than 0.05 was considered statistically significant. Statistical significance levels are denoted as follows: ns (not significant), $P > 0.05$; $*P < 0.05$; $**P < 0.01$; $***P < 0.001$; $****P < 0.0001$.

### Real-time quantitative RT-PCR analysis

DF-1 cells or HEK-293T cells were seeded in 6-well plates and then transfected with siRNA or plasmid, or infected with IBV at the indicated MOI, depending on the experimental requirements. At the specified time points, cells were transfected with poly I:C (20 μg/mL) or treated with IFNβ (1000 IU/mL) or TNFα (20 ng/mL). DMSO treatment was included in parallel experiments as a negative control. Total cellular RNA was extracted using Trizol reagent (Ambion, Austin, TX). cDNA was synthesized by reverse transcription using the EasyScript One-Step gDNA Removal and cDNA Synthesis SuperMix kit (Trans, AE311, China) with oligo dT primer. The cDNA served as a template for real-time qPCR using SYBR green master mix (Dongsheng Biotech, China) and corresponding primers. Real-time qPCR was conducted in the CFX-96 Bio-Rad instrument (Bio-Rad, USA), and the specificity of the amplified PCR products was confirmed by melting curve analysis after each reaction. The primers used to detect antiviral genes (IFNβ, IFITM3, and IL-8) and IBV replication/transcription (primers target gene 1 and primers target N gene) are listed in S3 Table.

Three confirmatory experiments were conducted, and the representative results were shown. Statistical analysis was performed using GraphPad Prism 8 software. The data are presented as bar graphs, with error bars representing the SD of three technical replicates within single experiment. For comparisons between two groups, significance was determined using a two-tailed independent Student's t-test. For comparisons among multiple groups (more than two), a one-way analysis of variance (ANOVA) followed by Tukey's post hoc test was used.

Statistical significance was denoted as follows: ns (not significant), P > 0.05; *P < 0.05; **P < 0.01; ***P < 0.001; ****P < 0.0001.

## Co-immunoprecipitation (Co-IP) and liquid chromatography-mass spectrometry

DF-1 cells or HEK-293T cells cultured in 6 cm plates were transfected with plasmid or infected with IBV. Cells were lysed using RIPA Lysis Buffer (Beyotime, P0013D, China) supplemented with 1 mM phenylmethylsulfonyl fluoride (PMSF) (Beyotime, ST506, China) and protease inhibitors (Millipore, USA). The cell lysates were centrifuged at 12,000 rpm for 15 min, and the supernatant proteins were incubated overnight at 4˚C with gentle rotation with 4 μg of IBV N antibody (mouse), or 2 μg of Flag Tag antibody (mouse) conjugated with Dynabeads Protein G Magnetic Beads (Invitrogen, 10004D, USA), or anti-HA magnetic Beads (Abmart, M20034, China). After incubation, the beads were washed three times with RIPA buffer and then precipitated using a magnetic stand. The beads were resuspended in 40 μL of RIPA lysis buffer and denatured by boiling at 100˚C for 5 min after adding 5 × SDS loading buffer (Beyotime, P0015L, China). Following centrifugation, the supernatants were subjected to Western blot analysis.

For the samples intended for Co-IP and liquid chromatography-mass spectrometry (LC-MS) analysis, HEK-293T cells were seeded in 10 cm plates and transfected with either Flag-tagged IBV N or the PXJ40 vector. At 30 h post-transfection, the cells were lysed, and Co-IP experiments were performed using an anti-Flag antibody. The Co-IP samples were then resolved by SDS-PAGE, and the gels were stained with Coomassie Brilliant Blue to visualize the proteins. The stained protein bands were excised from the gel, pooled, and subjected to protein identification via LC-MS analysis, which was conducted at Jingjie PTM BioLab (Hangzhou, China).

## Fluorescence in situ hybridization (FISH)

Cells were seeded onto chamber slides and transfected with Flag-tagged N proteins from various genera of coronaviruses for 24 h. Cells were fixed with 4% paraformaldehyde in DEPC-treated PBS for 15 min, permeabilized with 0.5% Triton X-100 for 15 min, and blocked with 3% BSA for 1 h, followed by endo-biotin blocking using a blocking kit (Invitrogen, #E21390) according to the manufacturer's instructions [129]. Cells were then incubated with the primary antibody anti-Flag for 1 h at 37˚C. Cells were washed three times with DEPC-treated PBS containing 0.2% Triton X-100. Cells were again fixed with 4% paraformaldehyde and washed three times with DEPC-treated PBS. Cells were then equilibrated in 2 × SSC (1 mg/mL t-RNA, 10% dextran sulfate, and 25% formamide) for 15 min at 42˚C, followed by hybridization of biotin-oligo d(T) (Promega, #Z5261) with the poly(A) tail of mRNA for approximately 12 h at 42˚C in a humid environment. Biotin-oligo d(T) (0.2 μmol/L) was diluted in DEPC-treated PBS containing 0.2% Triton X-100, 1 mM DTT, and 200 units/mL RNase inhibitor. After the hybridization step, samples were washed with 2 × SSC for 15 min and then with 0.5 × SSC for 15 min at 42˚C on a shaker. Cells were again fixed with 4% paraformaldehyde and washed with DEPC-treated PBS. Cells were then incubated with Alexa Fluor-conjugated secondary antibodies for 30 min, followed by FITC-conjugated streptavidin for 30 min at 37˚C. At the intervals of each step, cells were washed with DEPC-treated PBS containing 0.2% Triton X-100 three times. DAPI was then applied to stain the nuclei for 7 minutes. Cells were washed again three times and mounted onto glass slides using mounting reagent. Cells were examined using a Zeiss LSM880 confocal microscope.

The ImageJ program (NIH, USA) was used to quantify the intensities of the mRNA signals in vector-transfected and N protein-expressing cells. Following the manufacturer's

instructions, we analyzed three fields of view per sample to ensure the reliability and accuracy of our measurements. The data are presented as mean ± SD. Statistical analysis was performed using GraphPad Prism 8 software. Significance was determined using the two-tailed independent Student's t-test ($P < 0.05$) for comparisons between two groups. Statistical significance levels are denoted as follows: ns, $P > 0.05$; *, $P < 0.05$; **, $P < 0.01$; ***, $P < 0.001$; ****, $P < 0.0001$.

### Puromycin labelling

Puromycin resembles the 3′ end of tRNA and binds to growing peptide chains during translation, causing the cessation of protein synthesis and release of premature polypeptides containing puromycin [130]. Cells were cultured in four-well chamber slides and transfected with plasmids encoding N protein from various coronaviruses for 24 h. Following this, the cells were incubated with 5 μg/mL puromycin for 1 h at 37˚C. Cells were fixed for indirect immunofluorescence assay.

### Supporting information

**S1 Fig. IBV infection suppresses nuclear translocation of transcription factor IRF3, STAT2, and p65 in DF-1 cells.** (A-B) DF-1 cells were infected with IBV at an MOI of 1, followed by treatment with poly(I:C), IFNβ, or TNFα. Cells were harvested at the indicated time points and subjected to immunofluorescence analysis. Representative images from three independent experiments are shown. Scale bars: 10 μm.
(TIF)

**S2 Fig. IBV infection induces dislocation of Nups and Ran from nuclear envelope or nucleus to the cytoplasm in DF-1 cells.** (A-B) DF-1 cells were infected with IBV at an MOI of 1 or mock-infected, harvested at the indicated time points, and subjected to immunofluorescence analysis. Representative images from three independent experiments are shown. Scale bars: 10 μm.
(TIF)

**S3 Fig. IBV N protein alters the morphology of nuclear envelope and the ring signal of importin β1, and reduces the cytoplasmic signal of importin α1.** Vero cells were transfected with either the vector PXJ40 or a plasmid encoding IBV N protein. At 24 h post-transfection, cells were harvested and subjected to immunofluorescence analysis.
(TIF)

**S4 Fig. IBV encodes several proteins to counteract the host's innate immune response.** DF-1 cells were transfected with the PXJ40 vector or a plasmid encoding IBV proteins. At 24 h post-transfection, cells were transfected with poly(I:C) or treated with IFNβ or TNFα for 12 h, followed by qRT-PCR analysis. Statistical significance levels are denoted as follows: ns, $P > 0.05$; *$P < 0.05$; **$P < 0.01$; ***$P < 0.001$; ****$P < 0.0001$.
(TIF)

**S5 Fig. The N protein causes mRNA nuclear retention and shuts down protein translation.** (A) Vero cells were transfected with plasmids encoding Flag-tagged N protein from the indicated coronaviruses or the PXJ40 vector. After 24 h, mRNA was visualized using oligo dT probes (green) via *in situ* hybridization, followed by indirect immunofluorescence to detect the N protein (red). Nuclei were labeled with DAPI (blue). Representative images from three independent experiments are shown. The fluorescence signals of mRNA in vector transfected cells and N expressing cells from three fields of view were quantified using ImageJ. The

intensities of the fluorescence signals in the nucleus (black bars) and the cytoplasm (red bars) are presented as bar graphs. Error bars represent the SD. Statistical significance levels are denoted as follows: $^*P < 0.05$; $^{****}P < 0.0001$. (B) Vero cells were transfected with PXJ40 or with a plasmid encoding Flag-tagged N protein from the indicated coronaviruses. After 23 h, puromycin labeling (5 μg/mL) was performed for 1 h. Indirect immunofluorescence was then used to detect PABPC1 (red), N protein (magenta) and puromycin-labeled de novo synthesized peptides (green). Representative images from three independent experiments are shown. The relative fluorescence intensities of the IBV N protein and puromycin were quantified using ImageJ. The relative signal intensities and distributions of N and puromycin-labeled nascent peptides are shown as graph in the right panel.
(TIF)

**S6 Fig. Effect of PP1 and PP2A inhibitor okadaic acid treatment on phosphorylation of PKCα, PKCβ and NUP62, and induction of cytoplasmic dispersion of NUP62 and FG-Nups.** (A) DF-1 cells were treated with either DMSO or okadaic acid (1 μM) for 1 h and subjected to western blot analysis using the indicated antibodies. The intensities of p-PKCα, p-PKCβ, and p-NUP62 bands were normalized to total PKCα/β or NUP62. The ratio of p-PKCα, p-PKCβ, and p-NUP62 in okadaic acid-treated cells to DMSO-treated cells is denoted as p-PKCα (+:-), p-PKCβ (+:-), and p-NUP62 (+:-). (B) Vero cells were treated with either DMSO or okadaic acid (1 μM) for 1 h and subjected to immunostaining. Representative images from three independent experiments are shown, with scale bars indicating 10 μm. The fluorescence signals of Nup62 and FG-Nups in DMSO-treated and okadaic acid-treated cells were quantified from three fields of view using ImageJ. The intensities of the fluorescence signals in the nucleus (black bars) and the cytoplasm (red bars) are presented as bar graphs, with error bars representing the SD. Statistical significance is indicated as $^{****}$ for $P < 0.0001$ and $^{**}$ for $P < 0.01$.
(TIF)

**S7 Fig. In vitro interaction between IBV N protein and NUP62 or NUP42.** Plasmids encoding HA-tagged NUP62 or HA-tagged NUP42 were co-transfected with Flag-tagged IBV N protein or the PXJ40 control plasmid into HEK-293T cells for 24 h. The cell lysates were collected and subjected to Co-IP using an anti-Flag antibody to isolate the Flag-tagged N protein and any associated proteins. The immunoprecipitated complexes were then analyzed by Western blot analysis to detect the presence of HA-tagged NUP62 or HA-tagged NUP42, as well as the Flag-tagged IBV N protein.
(TIF)

**S1 Table. Dilution of primary antibodies and their cross-reactivity with corresponding chicken proteins.**
(DOCX)

**S2 Table. Primer sequences used for plasmid construction.**
(DOCX)

**S3 Table. Primer sequences used for real-time qPCR.**
(DOCX)

## Acknowledgments

We gratefully acknowledge Prof. Dingxiang Liu from South China Agricultural University, China, for providing the IBV Beaudette strain. Our sincere appreciation also goes to Prof.

Tongling Shan from the Shanghai Academy of Agricultural Sciences, CAAS, for generously sharing the plasmids encoding N protein from various genera of coronaviruses. Furthermore, we would like to thank Dr. Huan Wang, also affiliated with the Shanghai Academy of Agricultural Sciences, CAAS, for his contribution in generating the polyclonal IBV N antibody.

## Author Contributions

**Conceptualization:** Wenxiang Xue, Ying Liao.

**Formal analysis:** Wenxiang Xue, Ying Liao.

**Funding acquisition:** Chan Ding, Ying Liao.

**Investigation:** Wenxiang Xue, Hongyan Chu, Jiehuang Wang.

**Project administration:** Chan Ding, Ying Liao.

**Resources:** Yingjie Sun, Xusheng Qiu, Cuiping Song, Lei Tan.

**Supervision:** Yingjie Sun, Chan Ding, Ying Liao.

**Writing – original draft:** Wenxiang Xue, Ying Liao.

**Writing – review & editing:** Ying Liao.

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
