## [Decision Letter · Decision Letter 0]

10 Jun 2024

Dear Prof Liao,

Thank you very much for submitting your manuscript "Coronavirus Nucleocapsid Protein Enhances the binding of p-PKCα to RACK1: Implications for Inhibition of Nucleocytoplasmic Trafficking and Suppression of the Innate Immune Response" for consideration at PLOS Pathogens. As with all papers reviewed by the journal, your manuscript was reviewed by members of the editorial board and by several independent reviewers. In light of the reviews (below this email), we would like to invite the resubmission of a significantly-revised version that takes into account the reviewers' comments.

Dear Authors - Please kindly consider the Reviewers comments, addressing all of their concerns. Particularly pay attention to careful quantification and statistics for all data as well as the additional controls recommended by Reviewer 2. While the conclusions drawn may be valid it is not possible to draw these conclusions from a single representative image. Quantification of multiple cells must be included with relevant statistics and carefully worded methods into the number of samples/images/cells analyzed etc.

Additionally, some insight should be added on how this relates to real-world physiological animal infections and the larger impact of the work, also with relevance to existing literature. We find the conclusions interesting to the field, nonetheless, and look forward to seeing a revised manuscript.

We cannot make any decision about publication until we have seen the revised manuscript and your response to the reviewers' comments. Your revised manuscript is also likely to be sent to reviewers for further evaluation.

Sincerely,

Aaron T Irving, Ph.D.

Guest Editor

PLOS Pathogens

Ashley St. John

Section Editor

PLOS Pathogens

Michael Malim

Editor-in-Chief

PLOS Pathogens

orcid.org/0000-0002-7699-2064

Dear Authors - Please kindly consider the Reviewers comments, addressing all of their concerns. Particularly pay attention to careful quantification and statistics for all data as well as the additional controls recommended by Reviewer 2. While the conclusions drawn may be valid it is not possible to draw these conclusions from a single representative image. Quantification of multiple cells must be included with relevant statistics and carefully worded methods into the number of samples/images/cells analyzed etc.

Additionally, some insight should be added on how this relates to real-world physiological animal infections and the larger impact of the work, also with relevance to existing literature. We find the conclusions interesting to the field, nonetheless, and look forward to seeing a revised manuscript.

Reviewer's Responses to Questions

**Part I - Summary**

Reviewer #1: Overall, this manuscript presents a comprehensive investigation into the mechanisms by which the infectious bronchitis virus (IBV) interferes with the host's immune response through manipulation of nucleocytoplasmic trafficking. This study represents the first comprehensive report unveiling that coronaviruses harbor conserved mechanisms, orchestrated by the nucleocapsid protein, to manipulate nucleocytoplasmic trafficking and suppression of nuclear entry of multiple transcription factors. Additionally, it elucidates the pivotal role of the N-RACK1-PKC axis in Nup62 phosphorylation and disassembly of nuclear pore complex (NPC) integrity. Furthermore, this study provides direct evidence of the correlation between Nup phosphorylation and the NPC disassembly. It highlights the impact of coronavirus infection on NPC integrity and functionality, shedding light on novel mechanisms by which pan coronaviruses subverts host innate immune defense.

Reviewer #2: In this manuscript submitted to Plos Pathogens, the authors aimed to address the impact of coronavirus infection on the nucleocytoplasmic trafficking of host cells, focusing mainly on infection by the gamma-coronavirus, Infectious Bronchitis Coronavirus; IBV).

Previous studies already addressed how other coronaviruses (e.g., SARS-CoV and SARS-CoV-2) interfered with the nuclear transport system to suppress innate immunity (e.g., Frieman et al. J Virol. 2007 doi: 10.1128/JVI.01012-07; Miorin L et al. PNAS 2020 doi: 10.1073/pnas.2016650117 and Hall et al. PLoS Pathog. 2022 doi: 10.1371/journal.ppat.1010349). This inhibition involves in particular ORF6, which is absent in other coronaviruses, yet previous studies showed that IBV also inhibited IFNβ-mediated nuclear translocation of STAT1 (e.g., Kint et al. J Virol. 2015 doi: 10.1128/JVI.01057-15).

Here, the authors showed that IBV-induced cytoplasmic dispersion of several FG-Nups blocked the nuclear ingress of various transcription factors (i.e., IRF3, STAT1, STAT2 and, in part, p65 and p38 MAPK), and consequently the transcription of antiviral and pro-inflammatory genes. They further revealed that the IBV N protein contribute to this disturbance of the host nucleocytoplasmic trafficking by enhancing the association of p-PKCα with scaffold protein RACK1, leading to relocation of the p-PKCα-RACK1 complex to cytoplasm. Next, they propose that this event implied the phosphorylation and cytoplasmic dispersion of NUP62. Finally, they showed that the disruption of NPC integrity and interference with nuclear translocation of transcription factors by the N protein is conserved for different coronaviruses.

The topic and issues addressed are of great interest, as the modulation of nuclear translocation and antiviral responses was previously focused on the inhibition by ORF6, which is not conserved across different coronaviruses.

Despite the deep molecular analyses of this manuscript, for virtually all the conclusions on the perturbation of subcellular localization of regulators, only single representative imagings are presented. Therefore, the authors should include a quantification of these phenotypes. This main limitation along with a few others critic required additional experiments to better support the authors' conclusions, as suggested in these specific points.

Reviewer #3: The study uncovers a novel mechanism by which the coronavirus N protein interferes with the nuclear transport system, specifically through the interaction with RACK1 and p-PKCα, leading to the phosphorylation and cytoplasmic dispersion of NUP62. The research includes extensive immunofluorescence and biochemical analyses, providing robust evidence for the inhibitory effects of the IBV N protein on nuclear translocation of key transcription factors like IRF3, STAT1, STAT2, and p65. The findings demonstrate that the mechanism is conserved among N proteins from multiple coronaviruses, indicating a potentially universal strategy employed by these viruses to evade host immune responses. By elucidating a previously unknown mechanism of immune evasion employed by coronaviruses, the study provides critical insights that could inform the development of new therapeutic strategies targeting the NPC and related pathways to counteract coronavirus infections.

**Part II – Major Issues: Key Experiments Required for Acceptance**

Reviewer #1: Here are some review comments. Addressing these points would help strengthen the manuscript.

Major comments:

1. Improvements are needed in the presentation of immunostaining figures, as the target cells appear too small for clear visualization. Enlarging the target cells would enhance visibility.

2. Is the inhibition of nuclear entry by the N protein applicable to all types of nuclear localization signals (NLS)? Including reporter plasmids with NLS would indeed be beneficial for investigating the specificity of the inhibition of nuclear entry by the N protein.

3. Has the author investigated whether the N protein inhibits the nuclear export of host mRNA and subsequent protein translation?

4. The specific roles of PKCα and PKCβ in phosphorylating Nup62 were not addressed in this study. Is it feasible to individually knock down PKCα or PKCβ using specific siRNAs to investigate their respective contributions to Nup62 phosphorylation?

5. Based on this study, deleting the nuclear export signal (NES) of the N protein could affect its subcellular localization and function. Is it feasible to rescue a recombinant mutant virus by deleting the NES from the N protein? It is understood that in some cases, deleting essential viral elements like the NES may severely impair viral replication or assembly, making it difficult to rescue viable mutant viruses.

Reviewer #2: Major critics:

1- The imaging panels of Figure 1 clearly showed that IBV-mediated inhibition of nuclear translocation IRF3, STAT1 and STAT2, yet this is not readily observed for p65 and p38, as partly localized in the cytosol. The authors should thus reinforce their conclusions by a quantification of the cytosol versus nuclear subcellular localization of the different regulators for the tested conditions and presented as means for independent experiments. For example, this can be performed by quantifying their relative levels of colocalization with DAPI on the imaging analysis and/or by western blot analyses upon biochemical separation of nuclear and cytoplasm fractions.

2- Similar quantifications should be performed for NUP62, NUP153, NUP42, and TPR redistribution to the cytoplasm upon IBV infection shown in Figure 2 (especially since this observation is transient/vanish at late time-points for some regulators), as well as upon expression of the individual viral proteins (Figure 3, 4, 5, 8 and 12) and for Figure 7C and 9B-C.

3- In Figure 4D and 5E, to better conclude on the preferential regulation by N protein, the author should include similar RT-qPCR analyses of antiviral/pro-inflammatory responses upon expression of other viral proteins, as a comparison.

4- In Figure 9A, the change of phosphorylation levels of PKC and NUP62 are very limited in Vero cells as compared to DF1 cells. The authors should comment on this difference.

5- The impact of Enzastaurin inhibitor and down-regulation of PKC or RACK on viral replication is only revealed by analysis of the level of N protein in Figure 9D-E and 10A. Owing to the importance of these conclusions, the results should be confirmed by a different quantitative approach to demonstrate that the impact is on viral replication per se rather than the stability/expression levels of only the intracellular N protein.

6- In Figure 9F and 10B-C, the diminished impact of IBV infection on the antiviral response upon siRNA against PKC AND RACK can result - at least in part - from an indirect consequence of decreased viral replication and/or stability/expression levels of N protein in down-regulated cells (as mentioned in point 5) and in accordance with results of Figure 11F and 12 F upon individual expression of N protein. This question should be addressed.

Reviewer #3: 1 The study primarily relies on in vitro models, such as Vero and DF-1 cells, which might not fully replicate the complexity of an in vivo system.

2 The phosphorylation of IRF3, STAT1, p65 and p38 should be detected in figure 1, which may help to clarify the role of NPC in the suppression of IFNβ, IFITM3, and IL-8.

**Part III – Minor Issues: Editorial and Data Presentation Modifications**

Reviewer #1: Minor comments:

1. Background on coronaviruses in introduction section: The section detailing coronaviruses' genomic structure and pathogenicity provides essential context for understanding their impact on host cells and the immune response. However, it might be beneficial to briefly mention the zoonotic origins of some coronaviruses (e.g., SARS-CoV and SARS-CoV-2) to highlight their potential for cross-species transmission.

2. While the manuscript discusses the findings in the context of coronavirus infection, it could benefit from providing more background information on the significance of nucleocytoplasmic trafficking in antiviral immune responses. Offering a brief overview of previous studies or theories in this area would help readers better understand the significance of the current findings.

3. The transition from discussing general aspects of nucleocytoplasmic trafficking and coronavirus biology to introducing the specific objectives of the study could be smoother. Providing a brief overview or roadmap of the study's aims and methodologies would help orient the reader and set clear expectations for the subsequent sections.

4. The manuscript provides detailed experimental results, but there could be further discussion on the implications of these findings. For instance, what are the broader implications of IBV's interference with nucleocytoplasmic trafficking for our understanding of coronavirus pathogenesis and potential therapeutic interventions? Discussing these points would enrich the interpretation of the data.

5. Including graphical representations of key findings, such as graphs presenting quantitative data, could enhance the manuscript's visual appeal and aid in conveying complex concepts more effectively.

6. Exploring the size shift of Nup42 in future study could offer intriguing insights into its role in maintaining the integrity of the nuclear pore complex (NPC).

Reviewer #2: 1- In Figure 9A, the legend is completed: (+) is missing at 18 h.p.i..

2- Previously known function of the N protein in coronavirus life cycle and its impact on host cells should be introduced, especially in regards to its known ability to disrupt of host cell division (e.g., as reviewed by Su et al., Front Vet Sci. 2020 doi: 10.3389/fvets.2020.586826).

Reviewer #3: NO

PLOS authors have the option to publish the peer review history of their article (what does this mean?). If published, this will include your full peer review and any attached files.

Reviewer #1: No

Reviewer #2: No

Reviewer #3: No
---

## [Decision Letter · Decision Letter 1]

24 Oct 2024

PPATHOGENS-D-24-00486R1Coronavirus Nucleocapsid Protein Enhances the binding of p-PKCα to RACK1: Implications for Inhibition of Nucleocytoplasmic Trafficking and Suppression of the Innate Immune ResponsePLOS Pathogens Dear Dr. Liao, Thank you for submitting your manuscript to PLOS Pathogens. After careful consideration, we feel that it has merit but does not fully meet PLOS Pathogens's publication criteria as it currently stands. Therefore, we invite you to submit a revised version of the manuscript that addresses the points raised during the review process. Please submit your revised manuscript within 30 days Dec 23 2024 11:59PM. If you will need more time than this to complete your revisions, please reply to this message or contact the journal office at plospathogens@plos.org. Please include the following items when submitting your revised manuscript:*
A rebuttal letter that responds to each point raised by the editor and reviewer(s). You should upload this letter as a separate file labeled 'Response to Reviewers'. This file does not need to include responses to any formatting updates and technical items listed in the 'Journal Requirements' section below.*
A marked-up copy of your manuscript that highlights changes made to the original version. You should upload this as a separate file labeled 'Revised Manuscript with Track Changes'.*
An unmarked version of your revised paper without tracked changes. You should upload this as a separate file labeled 'Manuscript'. If you would like to make changes to your financial disclosure, competing interests statement, or data availability statement, please make these updates within the submission form at the time of resubmission. Guidelines for resubmitting your figure files are available below the reviewer comments at the end of this letter. We look forward to receiving your revised manuscript. Kind regards, Aaron T Irving, Ph.D.Guest EditorPLOS Pathogens Ashley St. JohnSection EditorPLOS Pathogens Michael Malim

Editor-in-Chief

PLOS Pathogens

orcid.org/0000-0002-7699-2064 **Journal Requirements:** **Additional Editor Comments (if provided):** Dear Xue and colleagues, we are happy to say the reviewers are largely satisfied with the changes to the manuscript and feel there is no major flaw. However, there are still some minor issues that require clarification prior to publication. Please read the reviewers comments carefully and address all of the minor concerns prior to resubmitting the final version.**Reviewers' Comments:** Reviewer's Responses to Questions

**Part I - Summary**

Reviewer #1: The relevant research of the manuscript findings unveil a novel, highly effective, and evolutionarily conserved.And further supplementation and modifications have been made under the suggestions and queries of the reviewers. mechanism.

Reviewer #2: In this manuscript re-submitted to Plos Pathogens, the authors adequately addressed my criticisms by including additional experiments, yet the following minor points should be addressed prior to publication.

Reviewer #4: Xue and colleagues investigated interference of IBV, a coronavirus infecting chickens, with nucleocytoplasmic trafficking of cellular transcription factors and establishment of innate antiviral responses. In brief, the authors show that IBV infection interferes with nuclear translocation of cellular transcription factors and expression of pro-inflammatory genes and genes involved in IFN responses. Furthermore, infection or directed expression of the N protein of IBV and other coronaviruses was found to alter posttranslational modification of NUP62 and other Nups and to compromise the integrity of the nuclear pore complex. In addition, the PKCα/β is shown to play a role in phosphorylation of NUP62, and RACK1 is demonstrated to be important for PKCα/β and NUP62 phosphorylation and inhibition of antiviral responses, with IBV and other coronavirus N proteins promoting p-PKCα interactions with RACK1. Finally, a nuclear export signal in IBV N protein is demonstrated to be required for suppression of innate immune responses. The results of a very comprehensive analysis are reported and the findings are of interest to the field.

**Part II – Major Issues: Key Experiments Required for Acceptance**

Reviewer #1: I have no new comments for revision.

Reviewer #2: NONE

Reviewer #4: None

**Part III – Minor Issues: Editorial and Data Presentation Modifications**

Reviewer #1: I have no new comments for revision.

Reviewer #2: 1- The revised S4 Fig revealed the inhibition by other viral proteins inhibit these signalings as well as now described by the authors. Nonetheless, their results also showed that some of the viral proteins potentiate the signaling induced by PolYIC, IFN-beta TFN treatment. The authors should comment on this observation.

2- The authors should better describe how the quantification of imaging was performed relative to the nuclear staining using DAPI, i.e., how the signal intensity is calculated the in regards to the limit of the nucleus - the relative area.

Reviewer #4: It is essential to state for all figure subpanels whether a single representative experiment or the average of several experiments is shown. If a single representative experiment is shown, please indicate how many confirmatory experiments were conducted. If the average is shown, please indicate how many experiments were averaged. Finally, please indicate whether error bars indicate SD or SEM.

The ability to interfere with Nup localization and STAT1 nuclear translocation was not only detected for the viral N protein but also for the E, M and 5a proteins. The authors night want to discuss in more detail the implications of this redundancy for IBV infection and for the specificity of the effects observed within the present study.

Occasional style issues show be remedied:

“economic animals” should read “farmed animals”.

“posing a continuous threaten in poultry farms” should read “posing a continuous threat to poultry farms”

“causing clinical symptoms such as vomiting and severe diarrhea, causing clinical symptoms such as vomiting and severe diarrhea” should read “causing clinical symptoms such as vomiting and severe diarrhea”

“to explore the shared mechanisms utilized by pan-coronaviruses” should read “to explore the shared mechanisms utilized by diverse coronaviruses”

“Initially, Vero cells, an IBV Beaudette strain adapted cell line,”. Presumably, the virus has been adapted to the cell line.

PLOS authors have the option to publish the peer review history of their article (what does this mean?). If published, this will include your full peer review and any attached files.

Reviewer #1: No

Reviewer #2: No

Reviewer #4: No

---

## [Editor Report · Decision Letter 2]

18 Nov 2024

Dear Prof Liao,

We are pleased to inform you that your manuscript 'Coronavirus Nucleocapsid Protein Enhances the binding of p-PKCα to RACK1: Implications for Inhibition of Nucleocytoplasmic Trafficking and Suppression of the Innate Immune Response' has been provisionally accepted for publication in PLOS Pathogens.

Best regards,

Aaron T Irving, Ph.D.

Guest Editor

PLOS Pathogens

Ashley St. John

Section Editor

PLOS Pathogens

Michael Malim

Editor-in-Chief

PLOS Pathogens

orcid.org/0000-0002-7699-2064

We are satisfied with all the minor revisions and are happy to accept your publication into PLOS Pathogens.
---

## [Editor Report · Acceptance letter]

24 Nov 2024

Dear Prof Liao,

We are delighted to inform you that your manuscript, " Coronavirus Nucleocapsid Protein Enhances the binding of p-PKCα to RACK1: Implications for Inhibition of Nucleocytoplasmic Trafficking and Suppression of the Innate Immune Response," has been formally accepted for publication in PLOS Pathogens.

Best regards,

Michael Malim

Editor-in-Chief

PLOS Pathogens

orcid.org/0000-0002-7699-2064